# MAML is a Noisy Contrastive Learner in Classification

**Chia-Hsiang Kao**[†]      **Wei-Chen Chiu**[†]      **Pin-Yu Chen**[‡]
[†]National Yang Ming Chiao Tung University, Taiwan      [‡]IBM Research
chkao.md04@nycu.edu.tw    walon@cs.nctu.edu.tw    pin-yu.chen@ibm.com

## Abstract

Model-agnostic meta-learning (MAML) is one of the most popular and widely adopted meta-learning algorithms, achieving remarkable success in various learning problems. Yet, with the unique design of nested inner-loop and outer-loop updates, which govern the task-specific and meta-model-centric learning, respectively, the underlying learning objective of MAML remains implicit, impeding a more straightforward understanding of it. In this paper, we provide a new perspective of the working mechanism of MAML. We discover that MAML is analogous to a meta-learner using a supervised contrastive objective in classification. The query features are pulled towards the support features of the same class and against those of different classes. Such contrastiveness is experimentally verified via an analysis based on the cosine similarity. Moreover, we reveal that vanilla MAML has an undesirable interference term originating from the random initialization and the cross-task interaction. We thus propose a simple but effective technique, the zeroing trick, to alleviate the interference. Extensive experiments are conducted on both mini-ImageNet and Omniglot datasets to validate the consistent improvement brought by our proposed method. [1]

## 1 Introduction

Humans can learn from very few samples. They can readily establish their cognition and understanding of novel tasks, environments, or domains even with very limited experience in the corresponding circumstances. *Meta-learning*, a subfield of machine learning, aims at equipping machines with such capacity to accommodate new scenarios effectively (Vilalta & Drissi, 2002; Grant et al., 2018). Machines learn to extract task-agnostic information so that their performance on unseen tasks can be improved (Hospedales et al., 2020).

One highly influential meta-learning algorithm is Model Agnostic Meta-Learning (MAML) (Finn et al., 2017), which has inspired numerous follow-up extensions (Nichol et al., 2018; Rajeswaran et al., 2019; Liu et al., 2019; Finn et al., 2019; Jamal & Qi, 2019; Javed & White, 2019). MAML estimates a set of model parameters such that an adaptation of the model to a new task only requires some updates to those parameters. We take the few-shot classification task as an example to review the algorithmic procedure of MAML. A few-shot classification problem refers to classifying samples from some classes (i.e. query data) after seeing a few examples per class (i.e. support data). In a meta-learning scenario, we consider a distribution of tasks, where each task is a few-shot classification problem and different tasks have different target classes. MAML aims to meta-train the base-model based on training tasks (i.e., the meta-training dataset) and evaluate the performance of the base-model on the testing tasks sampled from a held-out unseen dataset (i.e. the meta-testing dataset). In meta-training, MAML follows a bi-level optimization scheme composed of the inner loop and the outer loop, as shown in Appendix A (please refer to Section 2 for detailed definition). In the inner loop (also known as *fast adaptation*), the base-model $\theta$ is updated to $\theta'$ using the support set. In the outer loop, a loss is evaluated on $\theta'$ using the query set, and its gradient is computed with respect to $\theta$ to update the base-model. Since the outer loop requires computing the gradient of gradient (as the update in the inner loop is included in the entire computation graph), it is called second-order MAML (SOMAML). To prevent computing the Hessian matrix, Finn et al.

---
[1]Code available at https://github.com/IandRover/MAML_noisy_contrasive_learner

(2017) propose first-order MAML (FOMAML) that uses the gradient computed with respect to the inner-loop-updated parameters $\theta'$ to update the base-model.

The widely accepted intuition behind MAML is that the models are *encouraged* to learn general-purpose representations which are broadly applicable not only to the seen tasks but also to novel tasks (Finn et al., 2017; Raghu et al., 2020; Goldblum et al., 2020). Raghu et al. (2020) confirm this perspective by showing that during fast adaptation, the majority of changes being made are in the final linear layers. In contrast, the convolution layers (as the feature encoder) remain almost static. This implies that the models trained with MAML learn a good feature representation and that they only have to change the linear mapping from features to outputs during the fast adaptation. Similar ideas of freezing feature extractors during the inner loop have also been explored (Lee et al., 2019; Bertinetto et al., 2019; Liu et al., 2020), and have been held as an assumption in theoretical works (Du et al., 2021; Tripuraneni et al., 2020; Chua et al., 2021).

While this intuition sounds satisfactory, we step further and ask the following fundamental questions: (1) In what sense does MAML *guide* any model to learn general-purpose representations? (2) How do the inner and outer loops in the training mechanism of MAML collaboratively prompt to achieve so? (3) What is the role of support and query data, and how do they interact with each other? In this paper, we answer these questions and give new insights on the working mechanism of MAML, which turns out to be closely connected to supervised contrastive learning (SCL)[2].

Here, we provide a sketch of our analysis in Figure 1. We consider a setting of (a) a 5-way 1-shot paradigm of few-shot learning, (b) the mean square error (MSE) between the one-hot encoding of groundtruth label and the outputs as the objective function, and (c) MAML with a single inner-loop update. At the beginning of the inner loop, we set the linear layer $\mathbf{w}^0$ to zero. Then, the inner loop update of $\mathbf{w}^0$ is equivalent to adding the support features to $\mathbf{w}^0$. In the outer loop, the output of a query sample $q_1$ is actually the inner product between the query feature $\phi(q_1)$ and all support features (the learning rate is omitted for now). As the groundtruth is an one-hot vector, the encoder is trained to either minimize the inner product between the query features and the support features (when they are from different classes, as shown in the green box), or to pull the inner product between the query features and the support features to 1 (when they have the same label, as shown in the red box). Therefore, the inner loop and the outer loop together manifest a SCL objective. Particularly, as the vanilla implementation of MAML uses non-zero (random) initialization for the linear layer, we will show such initialization leads to a noisy SCL objective which would impede the training.

In this paper, we firstly review a formal definition of SCL, present a more general case of MAML with cross entropy loss in classification, and show the underlying learning protocol of vanilla MAML as an interfered SCL in Section 2. We then experimentally verify the supervised contrastiveness of MAML and propose to mitigate the interference with our simple but effective technique of the zero-initialization and zeroing trick (cf. Section 3). In summary, our main contributions are three-fold:

- We show MAML is implicitly an SCL algorithm in classification and the noise comes from the randomly initialized linear layer and the cross-task interaction.
- We verify the inherent contrastiveness of MAML based on the cosine similarity analysis.
- Our experiments show that applying the zeroing trick induces a notable improvement in testing accuracy during training and that that during meta-testing, a pronounced increase in the accuracy occurs when the zeroing trick is applied.

## 2 WHY MAML IS IMPLICITLY A NOISY SUPERVISED CONTRASTIVE ALGORITHM?

### 2.1 PRELIMINARY: SUPERVISED CONTRASTIVE LEARNING

In this work, we aim to bridge MAML and supervised contrastive learning (SCL) and attribute the success of MAML to SCL's capacity in learning good representations. Thus, we would like to introduce SCL briefly.

---

[2]We use the term *supervised contrastiveness* to refer to the setting of using ground truth label information to differentiate positive samples and negative samples (Khosla et al., 2020). This setting is different from (unsupervised/self-supervised) *contrastive learning*.

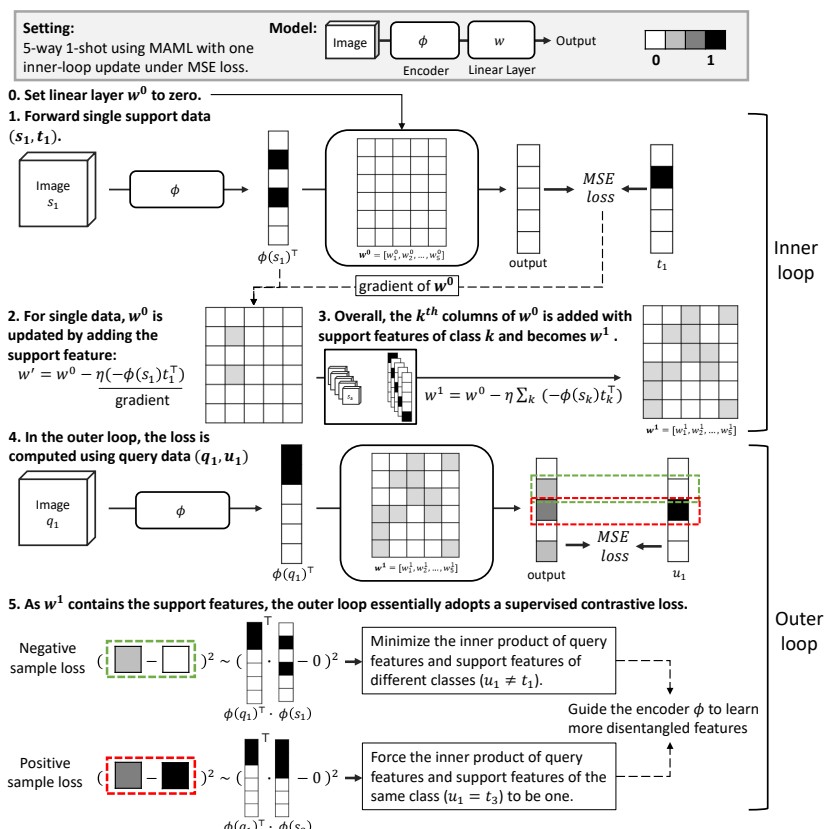

Figure 1: A step-by-step illustration showing the SCL objective underlying MAML. Assume the linear layer $\mathbf{w}^0$ to be zero, we find that, during the inner loop, the $i^{th}$ column of $\mathbf{w}^0$ is added with the support features of class $i$. In other words, the support features are memorized by the linear layer during the inner loop. In the outer loop, the output of a query sample is the inner product of $\phi(q_1)$ and $\mathbf{w}^1$, which is essentially the inner product of the query features and all the support features. The outer loop loss aims to minimize the MSE between the inner products and the one-hot label. Thus, MAML displays the characteristic of supervised contrastiveness. Besides, the support data acts as positive samples when the labels of support and query data match, and vice versa.

Supervised contrastive learning, proposed by Khosla et al. (2020), is a generalization of several metric learning algorithms, such as triplet loss and N-pair loss (Schroff et al., 2015; Sohn, 2016), and has shown the best performance in classification compared to SimCLR and CrossEntropy. In Khosla et al. (2020), SCL is described as "contrasts the set of all samples from the same class as positives against the negatives from the remainder of the batch" and "embeddings from the same class are pulled closer together than embeddings from different classes." For a sample $s$, the label information is leveraged to indicate positive samples (i.e., samples having the same label as sample $s$) and negative samples (i.e., samples having different labels to sample $s$). The loss of SCL is designed to increase the similarity (or decrease the metric distance) of embeddings of positive samples and to reduce the similarity (or increase the metric distance) of embeddings of negative samples (Khosla et al., 2020). In essence, SCL combines supervised learning and contrastive learning and differs from supervised learning in that the loss contains a measurement of the similarity (or distance) between the embedding of a sample and embeddings of its positive/negative sample pairs.

Now we give a formal definition of SCL. For a set of $N$ samples drawn from a $n$-class dataset. Let $i \in I = \{1, ..., N\}$ be the index of an arbitrary sample. Let $A(i) = I \setminus \{i\}$, $P(i)$ be the set of indices of all positive samples of sample $i$, and $N(i) = A(i) \setminus P(i)$ be the set of indices of all negative samples of sample $i$. Let $z_i$ indicates the embedding of sample $i$.

**Definition 1** *Let $M_{sim}$ be a measurement of similarity (e.g., inner product, cosine similarity). Training algorithms that adopt loss of the following form belong to SCL:*

$$L_{SCL} = \sum_i \sum_{p \in P(i)} c_{p,i}^- M_{sim}(z_i, z_p) + \sum_i \sum_{n \in N(i)} c_{n,i}^+ M_{sim}(z_i, z_n) + c \tag{1}$$

*where $c_{p,i}^- < 0$ and $c_{n,i}^+ > 0$ for all $n$, $p$ and $i$; and $c$ is a constant independent of samples.*

*We further define that a training algorithm that follows Eq.(1), but with either (a) $c_{n,i}^+ < 0$ for some $n, i$ or (b) $c$ is a constant dependent of samples, belongs to **noisy SCL**.*

## 2.2 Problem Setup

We provide the detailed derivation to show that MAML is implicitly a noisy SCL, where we adopt the few-shot classification as the example application. In this section, we focus on the meta-training period. Consider drawing a batch of tasks $\{T_1, \ldots, T_{N_{batch}}\}$ from a meta-training task distribution $D$. Each task $T_n$ contains a support set $S_n$ and a query set $Q_n$, where $S_n = \{(s_m, t_m)\}_{m=1}^{N_{way} \times N_{shot}}$, $Q_n = \{(q_m, u_m)\}_{m=1}^{N_{way} \times N_{query}}$, $s_m, q_m \in \mathbf{R}^{N_{in}}$ are data samples, and $t_m, u_m \in \{1, ..., N_{way}\}$ are labels. We denote $N_{way}$ the number of classes in each task, and $\{N_{shot}, N_{query}\}$ respectively the number of support and query samples per class. The architecture of our base-model comprises of a convolutional encoder $\phi : \mathbf{R}^{N_{in}} \to \mathbf{R}^{N_f}$ (parameterized by $\varphi$), a fully connected linear head $\mathbf{w} \in \mathbf{R}^{N_f \times N_{way}}$, and a Softmax output layer, where $N_f$ is the dimension of the feature space. We denote the $k^{th}$ column of $\mathbf{w}$ as $\mathbf{w_k}$. Note that the base-model parameters $\theta$ consist of $\varphi$ and $\mathbf{w}$.

As shown in Appendix A, both FOMAML and SOMAML adopt a training strategy comprising the inner loop and the outer loop. At the beginning of a meta-training iteration, we sample $N_{batch}$ tasks. For each task $T_n$, we perform inner loop updates using the inner loop loss (c.f. Eq. (2)) evaluated on the support data, and then evaluate the outer loop loss (c.f. Eq. (3)) on the updated base-model using the query data. In the $i^{th}$ step of the inner loop, the parameters $\{\varphi^{i-1}, \mathbf{w}^{i-1}\}$ are updated to $\{\varphi^i, \mathbf{w}^i\}$ using the multi-class cross entropy loss evaluated on the support dataset $S_n$ as

$$L_{\{\varphi^i, \mathbf{w}^i\}, S_n} = \mathop{\mathbf{E}}_{(s,t) \sim S_n} \sum_{j=1}^{N_{way}} \mathbf{1}_{j=t} [-\log \frac{\exp(\phi^i(s)^\top \mathbf{w_j}^i)}{\sum_{k=1}^{N_{way}} \exp(\phi^i(s)^\top \mathbf{w_k}^i)}] \tag{2}$$

After $N_{step}$ inner loop updates, we compute the outer loop loss using the query data $Q_n$:

$$L_{\{\varphi^{N_{step}}, \mathbf{w}^{N_{step}}\}, Q_n} = \mathop{\mathbf{E}}_{(q,u) \sim Q_n} [-\log \frac{\exp(\phi^{N_{step}}(q)^\top \mathbf{w_u}^{N_{step}})}{\sum_{k=1}^{N_{way}} \exp(\phi^{N_{step}}(q)^\top \mathbf{w_k}^{N_{step}})}] \tag{3}$$

Then, we sum up the outer loop losses of all tasks, and perform gradient descent to update the base-model's initial parameters $\{\varphi^0, \mathbf{w}^0\}$.

To show the supervised contrastiveness entailed in MAML, we adopt an assumption that *the Encoder $\phi$ is Frozen during the Inner Loop* (the **EFIL assumption**) and we discuss the validity of the assumption in Section 2.6. Without loss of generality, we consider training models with MAML with $N_{batch} = 1$ and $N_{step} = 1$, and we discuss the generalized version in Section 2.6. For simplicity, the $k^{th}$ element of model output $\frac{\exp(\phi(s)^\top \mathbf{w_k}^0)}{\sum_{j=1}^{N_{way}} \exp(\phi(s)^\top \mathbf{w_j}^0)}$ (respectively $\frac{\exp(\phi(q)^\top \mathbf{w_k}^1)}{\sum_{j=1}^{N_{way}} \exp(\phi(q)^\top \mathbf{w_j}^1)}$) of sample $s$ (respectively $q$) is denoted as $\mathrm{s}_k$ (respectively $\mathrm{q}_k$).

## 2.3 Inner Loop and Outer Loop Update of Linear Layer and Encoder

In this section, we primarily focus on the update of parameters in the case of FOMAML. The full derivation and discussion of SOMAML are provided in Appendix B.

**Inner loop update of the linear layer.** In the inner loop, the linear layer $\mathbf{w}^0$ is updated to $\mathbf{w}^1$ with a learning rate $\eta$ as shown in Eq. (4) in both FOMAML and SOMAML. In contrast to the example in Figure 1, the columns of the linear layer are added with the weighted sum of the features extracted from support samples (i.e., support features). Compared to $\mathbf{w_k}^0$, $\mathbf{w_k}^1$ is pushed towards the support features of the same class (i.e., class $k$) with strength of $1 - \mathrm{s}_k$, while being pulled away from the support features of different classes with strength of $\mathrm{s}_k$.

$$\mathbf{w_k}^1 = \mathbf{w_k}^0 - \eta \frac{\partial L_{\{\varphi, \mathbf{w}^0\}, S}}{\partial \mathbf{w_k}^0} = \mathbf{w_k}^0 + \eta \mathop{\mathbf{E}}_{(s,t) \sim S} (\mathbf{1}_{k=t} - \mathrm{s}_k) \phi(s) \tag{4}$$

**Outer loop update of the linear layer.** In the outer loop, $\mathbf{w}^0$ is updated using the query data with a learning rate $\rho$. For FOMAML, the final linear layer is updated as follows.

$$\mathbf{w_k'}^0 = \mathbf{w_k}^0 - \rho \frac{\partial L_{\{\varphi,\mathbf{w^1}\},Q}}{\partial \mathbf{w_k}^1} = \mathbf{w_k}^0 + \rho \mathop{\mathbf{E}}_{(q,u)\sim Q}(\mathbf{1}_{k=u} - q_k)\phi(q) \tag{5}$$

Note that the computation of $q_k$ requires the inner-loop updated $\mathbf{w}^1$. Generally speaking, Eq. (5) resembles Eq. (4). It is obvious that, in the outer loop, the query features are added weightedly to the linear layer, and the strength of change relates to the output value. In other words, after the outer loop update, the linear layer memorizes the query features of current tasks. This can cause a *cross-task interference* because in the next inner loop there would be additional inner products between the support features of the next tasks and the query features of the current tasks.

**Outer loop update of the encoder.** Using the chain rule, the gradient of the outer loop loss with respect to $\varphi$ (i.e., the parameters of the encoder) is given by $\frac{\partial L_{\{\varphi,\mathbf{w^1}\},Q}}{\partial \varphi} = \mathbf{E}_{(q,u)\sim Q} \frac{\partial L_{\{\varphi,\mathbf{w^1}\},Q}}{\partial \phi(q)} \frac{\partial \phi(q)}{\partial \varphi} + \mathbf{E}_{(s,t)\sim S} \frac{\partial L_{\{\varphi,\mathbf{w^1}\},Q}}{\partial \phi(s)} \frac{\partial \phi(s)}{\partial \varphi}$, where the second term can be neglected when FOMAML is considered. Below, we take a deeper look at the backpropagated error of one query data $(q,u) \sim Q$. The full derivation is provided in Appendix B.2.

$$\frac{\partial L_{\{\varphi,\mathbf{w^1}\},q}}{\partial \phi(q)} = \sum_{j=1}^{N_{way}} (q_j - \mathbf{1}_{j=u})\mathbf{w_j}^0 + \eta \mathop{\mathbf{E}}_{(s,t)\sim S}[-(\sum_{j=1}^{N_{way}} q_j s_j) + s_u + q_t - \mathbf{1}_{t=u}]\phi(s) \tag{6}$$

## 2.4 MAML IS A NOISY CONTRASTIVE LEARNER

**Reformulating the outer loop loss for the encoder as a noisy SCL loss.** We can observe from Eq. (6) that the actual loss for the encoder (evaluated on a single query data $(q,u) \sim Q$) is as the following.

$$L_{\{\varphi,\mathbf{w^1}\},q} = \sum_{j=1}^{N_{way}} \underbrace{(q_j - \mathbf{1}_{j=u})\mathbf{w_j}^{0\top}}_{\text{stop gradient}} \phi(q) + \eta \mathop{\mathbf{E}}_{(s,t)\sim S}[-\sum_{j=1}^{N_{way}} q_j s_j + s_u + q_t - \mathbf{1}_{t=u}]\underbrace{\phi(s)^{\top}}_{\text{stop gradient}} \phi(q) \tag{7}$$

For SOMAML, the range of "stop gradient" in the second term is different:

$$L_{\{\varphi,\mathbf{w^1}\},q} = \sum_{j=1}^{N_{way}} \underbrace{(q_j - \mathbf{1}_{j=u})\mathbf{w_j}^{0\top}}_{\text{stop gradient}} \phi(q) + \eta \mathop{\mathbf{E}}_{(s,t)\sim S}\underbrace{[-\sum_{j=1}^{N_{way}} q_j s_j + s_u + q_t - \mathbf{1}_{t=u}]\phi(s)^{\top}}_{\text{stop gradient}} \phi(q) \tag{8}$$

With these two reformulations, we observe the essential difference between FOMAML and SOMAML is the range of stop gradient. We provide detailed discussion and instinctive illustration in Appendix B.5 on how this explains the phenomenon that SOMAML often leads to faster convergence. To better deliberate the effect of each term in the reformulated outer loop loss, we define the first term in Eq. (7) or Eq. (8) as *interference term*, the second term as *noisy contrastive term*, and the coefficients $-\sum_{j=1}^{N_{way}} q_j s_j + s_u + q_t - \mathbf{1}_{t=u}$ as *contrastive coefficients*.

**Understanding the interference term.** In the case of $j = u$, the outer loop loss forces the model to minimize $(q_j - \mathbf{1})\mathbf{w_j}^{0\top}\phi(q)$. This can be problematic because (a) at the beginning of training, $\mathbf{w}^0$ is assigned with random values and (b) $\mathbf{w}^0$ is added with query features of previous tasks as shown in Eq. (5). Consequently, $\phi(q)$ is pushed to a direction composed of previous query features or to a random direction, introducing an unnecessary *cross-task interference* or an *initialization interference* that slows down the training of the encoder. Noting that the cross-task interference also occurs at the testing period, since, at testing stage, $\mathbf{w}^0$ is already added with query features of training tasks, which can be an interference to testing tasks.

**Understanding the noisy contrastive term.** When the query and support data have the same label (i.e., $u = t$), e.g., class 1, the contrastive coefficients becomes $-\sum_{j=2}^{N_{way}} q_j s_j - q_1 s_1 + s_1 + q_1 - \mathbf{1}$, which is $-\sum_{j=2}^{N_{way}} q_j s_j - (1 - q_1)(1 - s_1) < 0$. This indicates the encoder would be updated to maximize the inner product between $\phi(q)$ and the support features of the same class. However, when the query and support data are in different classes, the sign of the contrastive coefficient can

sometimes be negative. The outer loop loss thus cannot well contrast the query features against the support features of different classes, making this loss term not an ordinary SCL loss.

To better illustrate the influence of the interference term and the noisy contrastive term, we provide an ablation experiment in Appendix B.7. Theorem 1 below formally connects MAML to SCL.

**Theorem 1** *With the EFIL assumption, FOMAML is a noisy SCL algorithm. With assumptions of (a) EFIL and (b) a single inner-loop update, SOMAML is a noisy SCL algorithm.*

Proof: For FOMAML, both Eq. (7) (one inner loop update step) and Eq. (26) (multiple inner loop update steps) follows Definition 1. For SOMAML, Eq. (8) follows Definition 1.

**Introduction of the zeroing trick makes Eq.** (7) **and Eq.** (8) **SCL losses.** To tackle the interference term and make the contrastive coefficients more accurate, we introduce the zeroing trick: setting the $\mathbf{w}^0$ to be zero after each outer loop update, as shown in Appendix A. With the zeroing trick, the original outer loop loss (of FOMAML) becomes

$$L_{\{\varphi,\mathbf{w}^1\},q} = \eta \underset{(s,t)\sim S}{\mathbf{E}} \underbrace{(\mathbf{q}_t - \mathbf{1}_{\mathrm{t=u}})\phi(s)^\top}_{\text{stop gradient}} \phi(q) \tag{9}$$

For SOMAML, the original outer loop loss becomes

$$L_{\{\varphi,\mathbf{w}^1\},q} = \eta \underset{(s,t)\sim S}{\mathbf{E}} \underbrace{(\mathbf{q}_t - \mathbf{1}_{\mathrm{t=u}})}_{\text{stop gradient}} \phi(s)^\top \phi(q) \tag{10}$$

The zeroing trick brings two nontrivial effects: (a) eliminating the interference term in both Eq. (7) and Eq. (8); (b) making the contrastive coefficients follow SCL. For (b), since all the predictive values of support data become the same, i.e., $\mathbf{s}_k = \frac{1}{N_{way}}$, the contrastive coefficient becomes $\mathbf{q}_t - \mathbf{1}_{\mathrm{t=u}}$, which is negative when the support and query data have the same label, and positive otherwise. With the zeroing trick, the contrastive coefficient follows the SCL loss, as summarized below.

**Corollary 1** *With mild assumptions of (a) EFIL, (b) a single inner-loop update and (c) training with the zeroing trick (i.e., the linear layer is zeroed at the end of each outer loop), both FOMAML and SOMAML are SCL algorithms.*

Proof: Both Eq. (9) and Eq. (10) follow Definition 1.

The introduction of the zeroing trick makes the relationship between MAML and SCL more straightforward. Generally speaking, by connecting MAML and SCL, we can better understand other MAML-based meta-learning studies.

### 2.5 RESPONSES TO QUESTIONS IN SECTION 1

**In what sense does MAML guide any model to learn general-purpose representations?** Under the EFIL assumption, MAML is a noisy SCL algorithm in a classification paradigm. The effectiveness of MAML in enabling models to learn general-purpose representations can be attributed to the SCL characteristics of MAML.

**How do the inner and outer loops in the training mechanism of MAML collaboratively prompt to achieve so?** MAML adopts the inner and outer loops to perform noisy SCL sequentially. In the inner loop, the features of support data are memorized by $w$ via inner-loop update. In the outer loop, the softmax output of the query data thus contains the inner products between the support features and the query feature.

**What is the role of support and query data, and how do they interact with each other?** We show that the original loss in MAML can be reformulated as a loss term containing the inner products of the embedding of the support and query data. In FOMAML, the support features act as the reference, while the query features are updated to move towards the support features of the same class and against those of the different classes.

### 2.6 GENERALIZATION OF OUR ANALYSIS

In Appendix C, we provide the analysis where $N_{batch} \geq 1$ and $N_{step} \geq 1$. For the EFIL assumption, it can hardly be dropped because the behavior of the updated encoder is intractable. Besides, Raghu

et al. (2020) show that the representations of intermediate layers do not change notably during the inner loop of MAML, and thus it is understood that the main function of the inner loop is to change the final linear layer. Furthermore, the EFIL assumption is empirically reasonable, since previous works (Raghu et al., 2020; Lee et al., 2019; Bertinetto et al., 2019; Liu et al., 2020) yield comparable performance while leaving the encoder untouched during the inner loop.

With our analysis, one may notice that MAML is approximately a metric-based few-shot learning algorithm. From a high-level perspective, under the EFIL assumption, second-order MAML is similar to metric-based few-shot learning algorithms, such as MatchingNet (Vinyals et al., 2016), Prototypical network (Snell et al., 2017), and Relation network (Sung et al., 2018). The main difference lies in the metric and the way prototypes are constructed. Our work follows the setting adopted by MAML, such as using negative LogSoftmax as objective function, but we can effortlessly generalize our analysis to a MSE loss as had been shown in Figure 1. As a result, our work points out a new research direction in improving MAML by changing the objective functions in the inner and the outer loops, e.g., using MSE for the inner loop but negative LogSoftmax for the outer loop. Besides, in MAML, we often obtain the logits by multiplying the features by the linear weight $w$. Our work implies future direction as to alternatively substitute this inner product operation with other metrics or other similarity measurements such as cosine similarity or negative Euclidean distance.

## 3 EXPERIMENTAL RESULTS

In this section, we provide empirical evidence of the supervised contrastiveness of MAML and show that zero-initialization of $\mathbf{w}^0$, reduction in the initial norm of $\mathbf{w}^0$, or the application of zeroing trick can speed up the learning profile. This is applicable to both SOMAML and FOMAML.

### 3.1 SETUP

We conduct our experiments on the mini-ImageNet dataset (Vinyals et al., 2016; Ravi & Larochelle, 2017) and the Omniglot dataset (Lake et al., 2015). For the results on the Omniglot dataset, please refer to Appendix E. For the mini-ImageNet, it contains $84 \times 84$ RGB images of 100 classes from the ImageNet dataset with 600 samples per class. We split the dataset into 64, 16 and 20 classes for training, validation, and testing as proposed in (Ravi & Larochelle, 2017). We do not perform hyperparameter search and thus are not using the validation data. For all our experiments of applying MAML into few-shot classification problem, where we adopt two experimental settings: 5-way 1-shot and 5-way 5-shot, with the batch size $N_{batch}$ being 4 and 2, respectively (Finn et al., 2017). The few-shot classification accuracy is calculated by averaging the results over 400 tasks in the test phase. For model architecture, optimizer and other experimental details, please refer to Appendix D.1.

### 3.2 COSINE SIMILARITY ANALYSIS VERIFIES THE IMPLICIT CONTRASTIVENESS IN MAML

In Section 2, we show that the encoder is updated so that the query features are pushed towards the support features of the same class and pulled away from those of different classes. Here we verify this supervised contrastiveness experimentally. Consider a relatively overfitting scenario where there are five classes of images and for each class there are 20 support images and 20 query images. We fix the support and query set (i.e. the data is not resampled every iteration) to verify the concept that the support features work as positive and negative samples. Channel shuffling is used to avoid the undesirable channel memorization effect (Jamal & Qi, 2019; Rajendran et al., 2020). We train the model using FOMAML and examine how well the encoder can separate the data of different classes in the feature space by measuring the averaged cosine similarities between the features of each class. The results are averaged over 10 random seeds.

As shown in the top row of Figure 2, the model trained with MAML learns to separate the features of different classes. Moreover, the contrast between the diagonal and the off-diagonal entries of the heatmap increases as we remove the initialization interference (by zero-initializing $\mathbf{w}^0$, shown in the middle row) and remove the cross-task interference (by applying the zeroing trick, shown in the bottom row). The result agrees with our analysis that MAML implicitly contains the interference term which can impede the encoder from learning a good feature representation. For experiments on semantically similar classes of images, the result is shown in Section D.3.

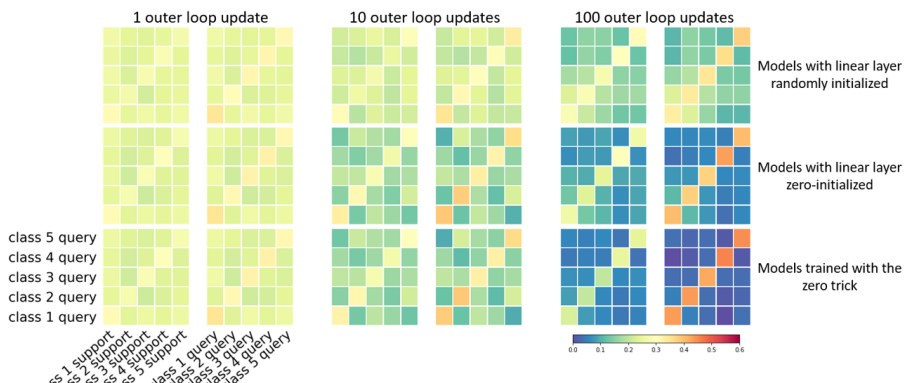

Figure 2: **The supervised contrastiveness entailed in MAML manifests when zero initialization or the zeroing trick is applied.** The value in the heatmap is calculated by averaging the pairwise cosine similarities between query features or between query features and support features. We consider the setting of having randomly initialized linear layer (top), zero-initialized linear layer (middle), and the zeroing trick (bottom), and experiment with various numbers of outer loop updates. The readers are encouraged to compare the results between different rows.

### 3.3 ZEROING LINEAR LAYER AT TESTING TIME INCREASES TESTING ACCURACY

Before starting our analysis on benchmark datasets, we note that the cross-task interference can also occur during meta-testing. In the meta-testing stage, the base-model is updated in the inner loop using support data $S$ and then the performance is evaluated using query data $Q$, where $S$ and $Q$ are drawn from a held-out, unseen meta-testing dataset. Recall that at the end of the outer loop (in meta-training stage), the query features are added weightedly to the linear layer $\mathbf{w}^0$. In other words, at the beginning of meta-testing, $\mathbf{w}^0$ is already added with the query features of previous training tasks, which can drastically influence the performance on the unseen tasks.

To validate this idea, we apply the zeroing trick at meta-testing time (which we refer to zeroing $\mathbf{w}^0$ at the beginning of the meta-testing time) and show such trick increases the testing accuracy of the model trained with FOMAML. As illustrated in Figure 3, compared to directly entering meta-testing (i.e. the subplot at the left), additionally zeroing the linear layer at the beginning of each meta-testing time (i.e. the subplot at the right) increases the testing accuracy of the model whose linear layer is randomly initialized or zero-initialized (denoted by the red and orange curves, respectively). And the difference in testing performance sustains across the whole training session.

In the following experiments, we evaluate the testing performance only with zeroing the linear layer at the beginning of the meta-testing stage. By zeroing the linear layer, the potential interference brought by the prior (of the linear layer) is ignored. Then, we can fully focus on the capacity of the encoder in learning a good feature representation.

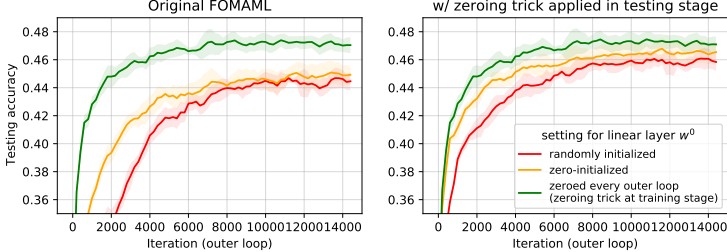

Figure 3: **Using the zeroing trick at meta-testing stage improves performance.** The left/right subplot shows the performance of models without/with $\mathbf{w}^0$ zeroed at the beginning of meta-testing time. The curves in red: $\mathbf{w}^0$ is randomly initialized. The curves in yellow: $\mathbf{w}^0$ is zeroed at initialization. The curves in green: the models trained with training trick applied in the training stage.

### 3.4 SINGLE INNER LOOP UPDATE SUFFICES WHEN USING THE ZEROING TRICK

In Eq. (4) and Eq. (21), we show that the features of the support data are added to the linear layer in the inner loop. Larger number of inner loop update steps can better offset the effect of interference brought by a non-zeroed linear layer. In other words, when the models are trained with the zeroing trick, a larger number of inner loop updates can not bring any benefit. We validate this intuition in Figure 4 under a 5-way 1-shot setting. In the original FOMAML, the models trained with a single inner loop update step (denoted as red curve) converge slower than those trained with update step of 7 (denoted as purple curve). On the contrary, when the models are trained with the zeroing trick, models with various inner loop update steps converge at the same speed.

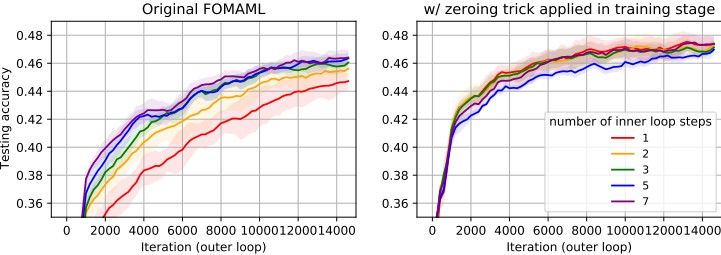

Figure 4: **With the zeroing trick, a larger number of inner loop update steps is not necessary.** In original MAML, a larger number of inner loop update steps is preferred as it generally yields better testing accuracy even with zeroing trick applied in the meta-testing stage (refer to the left figure). However, models trained using the zeroing trick do not show this trend (refer to the right figure).

### 3.5 EFFECT OF INITIALIZATION AND THE ZEROING TRICK

In Eq. (7), we observe an interference derived from the historical task features or random initialization. We validate our formula by examining the effects of (1) reducing the norm of $\mathbf{w}^0$ at initialization and (2) applying the zeroing trick. From Figure 5, the performance is higher when the initial norm of $\mathbf{w}^0$ is lower. Compared to random initialization, reducing the norm via down-scaling $\mathbf{w}^0$ by $0.7$ yields visible differences. Besides, the testing accuracy of MAML with zeroing trick (the purple curve) outperforms that of original MAML.

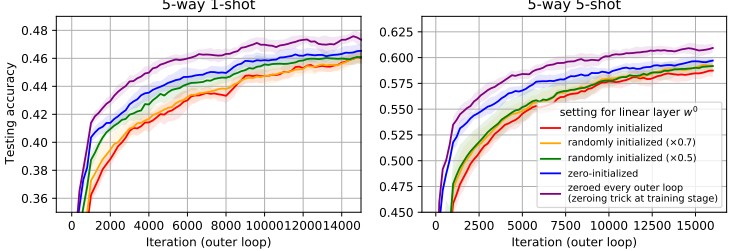

Figure 5: **Effect of initialization and the zeroing trick on testing performance.** Both reducing the norm of $\mathbf{w}^0$ and zeroing $\mathbf{w}^0$ each outer loop (i.e., the zeroing trick) increase the testing accuracy. The curves in red: models with $\mathbf{w}^0$ randomly initialized. The curves in orange/green: reducing the value of $\mathbf{w}^0$ at initialization by a factor of $0.7$/ $0.5$. The curve in blue: $\mathbf{w}^0$ is zero-initialized. The curve in blue: models trained with the zeroing trick.

## 4 CONCLUSION

This paper presents an extensive study to demystify how the seminal MAML algorithm guides the encoder to learn a general-purpose feature representation and how support and query data interact. Our analysis shows that MAML is implicitly a supervised contrastive learner using the support features as positive and negative samples to direct the update of the encoder. Moreover, we unveil an interference term hidden in MAML originated from the random initialization or cross-task interaction, which can impede the representation learning. Driven by our analysis, removing the interference term by a simple zeroing trick renders the model unbiased to seen or unseen tasks. Furthermore, we show constant improvements in the training and testing profiles with this zeroing trick, with experiments conducted on the mini-ImageNet and Omniglot datasets.

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

APPENDIX

## A  ORIGINAL MAML AND MAML WITH THE ZEROING TRICK

---

**Algorithm 1** Second-order MAML

---

    **Require**: Task distribution $D$
    **Require**: $\eta, \rho$: inner loop and outer loop learning rates
    **Require**: Randomly initialized base-model parameters $\theta$
1: **while** not done **do**
2:    Sample tasks $\{T_1, \ldots T_{N_{batch}}\}$ from $D$
3:    **for** $n = 1, 2, \ldots, N_{batch}$ **do**
4:       $\{S_n, Q_n\} \leftarrow$ sample from $T_n$
5:       $\theta_n = \theta$
6:       **for** $i = 1, 2, \ldots, N_{step}$ **do**
7:          $\theta_n \leftarrow \theta_n - \eta \nabla_{\theta_n} L_{\theta_n, S_n}$
8:       **end for**
9:    **end for**
10:   Update $\theta \leftarrow \theta - \rho \sum_{n=1}^{N_{batch}} \nabla_\theta L_{\theta_n, Q_n}$
11: **end while**

---

**Algorithm 2** First-order MAML

---

    **Require**: Task distribution $D$
    **Require**: $\eta, \rho$: inner loop and outer loop learning rates
    **Require**: Randomly initialized base-model parameters $\theta$
1: **while** not done **do**
2:    Sample tasks $\{T_1, \ldots T_{N_{batch}}\}$ from $D$
3:    **for** $n = 1, 2, \ldots, N_{batch}$ **do**
4:       $\{S_n, Q_n\} \leftarrow$ sample from $T_n$
5:       $\theta_n = \theta$
6:       **for** $i = 1, 2, \ldots, N_{step}$ **do**
7:          $\theta_n \leftarrow \theta_n - \eta \nabla_{\theta_n} L_{\theta_n, S_n}$
8:       **end for**
9:    **end for**
10:   Update $\theta \leftarrow \theta - \rho \sum_{n=1}^{N_{batch}} \nabla_{\theta_n} L_{\theta_n, Q_n}$
11: **end while**

---

**Algorithm 3** Second-order MAML with the zeroing trick

---

    **Require**: Task distribution $D$
    **Require**: $\eta, \rho$: inner loop and outer loop learning rates
    **Require**: Randomly initialized base-model parameters $\theta$
1: Set $\mathbf{w} \leftarrow 0$ (the zeroing trick)
2: **while** not done **do**
3:    Sample tasks $\{T_1, \ldots T_{N_{batch}}\}$ from $D$
4:    **for** $n = 1, 2, \ldots, N_{batch}$ **do**
5:       $\{S_n, Q_n\} \leftarrow$ sample from $T_n$
6:       $\theta_n = \theta$
7:       **for** $i = 1, 2, \ldots, N_{step}$ **do**
8:          $\theta_n \leftarrow \theta_n - \eta \nabla_{\theta_n} L_{\theta_n, S_n}$
9:       **end for**
10:   **end for**
11:   Update $\theta \leftarrow \theta - \rho \sum_{n=1}^{N_{batch}} \nabla_\theta L_{\theta_n, Q_n}$
12:   Set $\mathbf{w} \leftarrow 0$ (the zeroing trick)
13: **end while**

---

## B  Supplementary Derivation

In this section, we provide the full generalization and further discussion that supplement the main paper. We consider the case of $N_{batch} = 1$ and $N_{step} = 1$ under the EFIL assumption. We provide the outer loop update of the linear layer under SOMAML in Section B.1. Next, we offer the full derivation of the outer loop update of the encoder in Section B.2. Then, we reformulate the outer loop loss for the encoder in both FOMAML and SOMAML in Section B.3 and Section B.4. Afterward, we discuss the main difference in FOMAML and SOMAML in detail in Section B.5. Finally, we show the performance of the models trained using the reformulated loss in Section B.6.

### B.1  The Derivation of Outer Loop Update for the Linear Layer Using SOMAML

Here, we provide the complete derivation of the outer loop update for the linear layer. Using SO-MAML with support set $S$ and query set $Q$, the update of the linear layer follows

$$
\begin{aligned}
\mathbf{w_k'}^{\mathbf{0}} = \mathbf{w_k}^0 - \rho \frac{\partial L_{\{\varphi, \mathbf{w^1}\}, Q}}{\partial \mathbf{w_k}^0} &= \mathbf{w_k}^0 - \rho \sum_{m=1}^{N_{way}} \frac{\partial \mathbf{w_m}^1}{\partial \mathbf{w_k}^0} \cdot \frac{\partial L_{\{\varphi, \mathbf{w^1}\}, Q}}{\partial \mathbf{w_m}^1} \\
&= \mathbf{w_k}^0 - \rho \frac{\partial \mathbf{w_k}^1}{\partial \mathbf{w_k}^0} \cdot \frac{\partial L_{\{\varphi, \mathbf{w^1}\}, Q}}{\partial \mathbf{w_k}^1} - \rho \sum_{m \neq k}^{N_{way}} \frac{\partial \mathbf{w_m}^1}{\partial \mathbf{w_k}^0} \cdot \frac{\partial L_{\{\varphi, \mathbf{w^1}\}, Q}}{\partial \mathbf{w_m}^1} \\
&= \mathbf{w_k}^0 + \rho[I - \eta \operatorname*{\mathbf{E}}_{(s,t) \sim S} (\mathsf{s}_k - \mathsf{s}_k^2) \phi(s) \phi(s)^T] \operatorname*{\mathbf{E}}_{(q,u) \sim Q} (\mathbf{1}_{\mathrm{k=u}} - \mathsf{q}_k) \phi(q) \\
&\quad + \rho \eta \sum_{m \neq k} [\operatorname*{\mathbf{E}}_{(s,t) \sim S} (\mathsf{s}_m \mathsf{s}_k) \phi(s) \phi(s)^T][\operatorname*{\mathbf{E}}_{(q,u) \sim Q} (\mathbf{1}_{\mathrm{m=u}} - \mathsf{q}_m) \phi(q)] \\
&= \mathbf{w_k}^0 + \rho[I - \eta \operatorname*{\mathbf{E}}_{(s,t) \sim S} \mathsf{s}_k \phi(s) \phi(s)^T] \operatorname*{\mathbf{E}}_{(q,u) \sim Q} (\mathbf{1}_{\mathrm{k=u}} - \mathsf{q}_k) \phi(q) \\
&\quad + \rho \eta \sum_{m=1}^{N_{way}} [\operatorname*{\mathbf{E}}_{(s,t) \sim S} (\mathsf{s}_m \mathsf{s}_k) \phi(s) \phi(s)^T][\operatorname*{\mathbf{E}}_{(q,u) \sim Q} (\mathbf{1}_{\mathrm{m=u}} - \mathsf{q}_m) \phi(q)]
\end{aligned}
\tag{11}
$$

We can further simplify Eq. (11) to Eq. (12) with the help of the zeroing trick.

$$
\mathbf{w_k'}^{\mathbf{0}} = \rho[I - \eta \operatorname*{\mathbf{E}}_{(s,t) \sim S} \mathsf{s}_k \phi(s) \phi(s)^T] \operatorname*{\mathbf{E}}_{(q,u) \sim Q} (\mathbf{1}_{\mathrm{k=u}} - \mathsf{q}_k) \phi(q)
\tag{12}
$$

This is because the zeroing trick essentially turns the logits of all support samples to zero, and consequently the predicted probability (softmax) output $\mathsf{s}_m$ becomes $\frac{1}{N_{way}}$ for all channel $m$. Therefore, the third term in Eq. (11) turns out to be zero (c.f. Eq. (13)). The equality of Eq. (13) holds since the summation of the (softmax) outputs is one.

$$
\begin{aligned}
\frac{\rho \eta}{N_{way}^2} \sum_{m=1}^{N_{way}} [\operatorname*{\mathbf{E}}_{(s,t) \sim S} \phi(s) \phi(s)^T][\operatorname*{\mathbf{E}}_{(q,u) \sim Q} (\mathbf{1}_{\mathrm{m=u}} - \mathsf{q}_m) \phi(q)] \\
= \frac{\rho \eta}{N_{way}^2} [\operatorname*{\mathbf{E}}_{(s,t) \sim S} \phi(s) \phi(s)^T] \operatorname*{\mathbf{E}}_{(q,u) \sim Q} \phi(q) \sum_{m=1}^{N_{way}} (\mathbf{1}_{\mathrm{m=u}} - \mathsf{q}_m) = 0
\end{aligned}
\tag{13}
$$

### B.2  The Full Derivation of the Outer Loop Update of the Encoder.

As the encoder $\phi$ is parameterized by $\varphi$, the outer loop gradient with respect to $\varphi$ is given by $\frac{\partial L_{\{\varphi, \mathbf{w^1}\}, Q}}{\partial \varphi} = \mathbf{E}_{(q,u) \sim Q} \frac{\partial L_{\{\varphi, \mathbf{w^1}\}, Q}}{\partial \phi(q)} \frac{\partial \phi(q)}{\partial \varphi} + \mathbf{E}_{(s,t) \sim S} \frac{\partial L_{\{\varphi, \mathbf{w^1}\}, Q}}{\partial \phi(s)} \frac{\partial \phi(s)}{\partial \varphi}$. We take a deeper look at the backpropagated error $\frac{\partial L_{\{\varphi, \mathbf{w^1}\}, Q}}{\partial \phi(q)}$ of the feature of one query data $(q, u) \sim Q$, based on the following

form:

$$
\begin{aligned}
-\frac{\partial L_{\{\varphi,\mathbf{w^1}\},Q}}{\partial \phi(q)} &= \mathbf{w_u}^1 - \sum_{j=1}^{N_{way}} (\mathbf{q}_j \mathbf{w_j}^1) = \sum_{j=1}^{N_{way}} (\mathbf{1}_{\mathrm{j=u}} - \mathbf{q}_j)\mathbf{w_j}^1 \\
&= \sum_{j=1}^{N_{way}} (\mathbf{1}_{\mathrm{j=u}} - \mathbf{q}_j)\mathbf{w_j}^0 + \eta \sum_{j=1}^{N_{way}} [\mathbf{1}_{\mathrm{j=u}} - \mathbf{q}_j][\mathop{\mathbf{E}}_{(s,t)\sim S} (\mathbf{1}_{\mathrm{j=t}} - \mathbf{s}_j)\phi(s)] \\
&= \sum_{j=1}^{N_{way}} (\mathbf{1}_{\mathrm{j=u}} - \mathbf{q}_j)\mathbf{w_j}^0 + \eta \mathop{\mathbf{E}}_{(s,t)\sim S} [(\sum_{j=1}^{N_{way}} \mathbf{q}_j \mathbf{s}_j) - \mathbf{s}_u - \mathbf{q}_t + \mathbf{1}_{\mathrm{t=u}}]\phi(s)
\end{aligned}
\tag{14}
$$

## B.3 REFORMULATION OF THE OUTER LOOP LOSS FOR THE ENCODER AS NOISY SCL LOSS.

We can derive the actual loss (evaluated on a single query data $(q, u) \sim Q$) that the encoder uses under FOMAML scheme as follows:

$$
L_{\{\varphi,\mathbf{w^1}\},q} = \sum_{j=1}^{N_{way}} \underbrace{(\mathbf{q}_j - \mathbf{1}_{\mathrm{j=u}})\mathbf{w_j}^{0\top}}_{\text{stop gradient}} \phi(q) - \eta \mathop{\mathbf{E}}_{(s,t)\sim S} \underbrace{[(\sum_{j=1}^{N_{way}} \mathbf{q}_j \mathbf{s}_j) - \mathbf{s}_u - \mathbf{q}_t + \mathbf{1}_{\mathrm{t=u}}]\phi(s)^\top}_{\text{stop gradient}} \phi(q)
\tag{15}
$$

For SOMAML, we need to additionally plug Eq. (4) into Eq. (3).

$$
L_{\{\varphi,\mathbf{w^1}\},q} = \sum_{j=1}^{N_{way}} \underbrace{(\mathbf{q}_j - \mathbf{1}_{\mathrm{j=u}})\mathbf{w_j}^{0\top}}_{\text{stop gradient}} \phi(q) - \eta \mathop{\mathbf{E}}_{(s,t)\sim S} [(\sum_{j=1}^{N_{way}} \mathbf{q}_j \mathbf{s}_j) - \mathbf{s}_u - \mathbf{q}_t + \mathbf{1}_{\mathrm{t=u}}] \underbrace{\phi(s)^\top}_{\text{stop gradient}} \phi(q)
\tag{16}
$$

## B.4 INTRODUCTION OF THE ZEROING TRICK MAKES EQ. (7) AND EQ. (8) SCL LOSSES.

Apply the zeroing trick to Eq. (7) and Eq. (8), we can derive the actual loss Eq. (17) and Eq. (18) that the encoder follows.

$$
L_{\{\varphi,\mathbf{w^1}\},q} = \eta \mathop{\mathbf{E}}_{(s,t)\sim S} \underbrace{(\mathbf{q}_t - \mathbf{1}_{\mathrm{t=u}})\phi(s)^\top}_{\text{stop gradient}} \phi(q)
\tag{17}
$$

$$
L_{\{\varphi,\mathbf{w^1}\},q} = \eta \mathop{\mathbf{E}}_{(s,t)\sim S} \underbrace{(\mathbf{q}_t - \mathbf{1}_{\mathrm{t=u}})}_{\text{stop gradient}} \phi(s)^\top \phi(q)
\tag{18}
$$

With these two equations, we can observe the essential difference in FOMAML and SOMAML is the range of stopping gradients. We would further discuss the implication of different ranges of gradient stopping in Appendix B.5.

## B.5 DISCUSSION ABOUT THE DIFFERENCE BETWEEN FOMAML AND SOMAML

Central to the mystery of MAML is the difference between FOMAML and SOMAML. Plenty of work is dedicated to approximating or estimating the second-order derivatives in the MAML algorithm in a more computational-efficient or accurate manner (Song et al., 2020; Rothfuss et al., 2019; Liu et al., 2019). With the EFIL assumption and our analysis through connecting SCL to these algorithms, we found that we can better understand the distinction between FOMAML and SOMAML from a novel perspective. To better understand the difference, we can compare Eq. (7) with Eq. (8) or compare Eq. (9) with Eq. (10). To avoid being distracted by the interference terms, we provide the analysis of the latter.

The main difference between Eq. (9) and Eq. (10) is the range of gradient stopping and we will show that this difference results in a significant distinction in the feature space. To begin with, by chain rule, we have $\frac{\partial L}{\partial \varphi} = \mathbf{E}_{(q,u)\sim Q} \frac{\partial L}{\partial \phi(q)} \frac{\partial \phi(q)}{\partial \varphi} + \mathbf{E}_{(s,t)\sim S} \frac{\partial L}{\partial \phi(s)} \frac{\partial \phi(s)}{\partial \varphi}$. As we specifically want to know how the encoded features are updated given different losses, we can look at the terms $\frac{\partial L}{\partial \phi(q)}$ and $\frac{\partial L}{\partial \phi(s)}$ by differentiating Eq. (9) and Eq. (10) with respect to the features of query data $q$ and support data $s$, respectively.

FOMAML:

$$\frac{\partial L}{\partial \phi(q)} = \eta \mathop{\mathbf{E}}_{(s,t) \sim S} (\mathbf{q}_t - \mathbf{1}_{t=u}) \phi(s)$$
$$\frac{\partial L}{\partial \phi(s)} = 0 \tag{19}$$

SOMAML:

$$\frac{\partial L}{\partial \phi(q)} = \eta \mathop{\mathbf{E}}_{(s,t) \sim S} (\mathbf{q}_t - \mathbf{1}_{t=u}) \phi(s)$$
$$\frac{\partial L}{\partial \phi(s)} = \eta \mathop{\mathbf{E}}_{(s,t) \sim S} (\mathbf{q}_t - \mathbf{1}_{t=u}) \phi(q) \tag{20}$$

Obviously, as the second equation in Eq (19) is zero, we know that in FOMAML, the update of the encoder does consider the change of the support features. The encoder is updated to move the query features closer to support features of the same class and further to support features of different classes in FOMAML. On the contrary, we can tell from the above equations that in SOMAML, the encoder is updated to make support features and query features closer if both come from the same class and make support features and query features further if they come from different classes.

We illustrate the difference in Figure 6. For simplicity, we do not consider the scale of the coefficients but their signs. The subplot on the left indicates that this FOMAML loss guides the encoder to be updated so that the feature of the query data moves 1) towards the support feature of the same class, and 2) against the support features of the different classes. On the other hand, the SOMAML loss guides the encoder to be updated so that 1) when the support data and query data belong to the same class, their features move closer, and otherwise, their features move further. This generally explains why models trained using SOMAML generally converge faster than those trained using FOMAML.

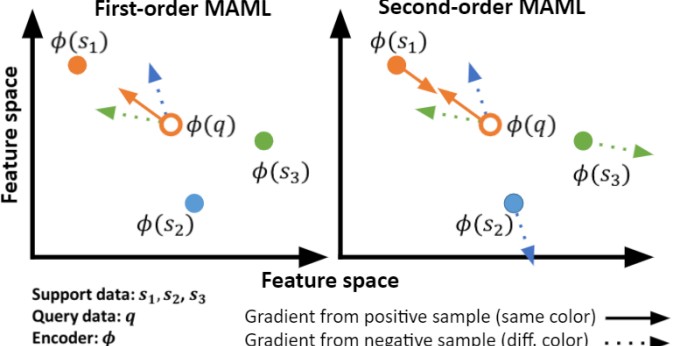

Figure 6: **Illustration of the distinction of FOMAML and SOMAML**. Conceptually speaking, the objective function of FOMAML aims to change the features of the query data; in contrast, that of SOMAML seeks to change the query's features and support data simultaneously. In this figure, the support data and query data features are plotted as solid and hollow circles, respectively. The different colors represent different classes. The solid and hollow arrows indicate the gradient calculated from positive and negative samples, respectively. Note that this is in the feature space, not the weight space.

B.6 EXPLICITLY COMPUTING THE REFORMULATING LOSS USING EQ. (7) AND EQ. (8)

Under the EFIL assumption, we show that MAML can be reformulated as a loss taking noisy SCL form. Below, we consider a setting of 5-way 1-shot mini-ImageNet few-shot classification task, under the condition of no inner loop update of the encoder. (This is the assumption that our derivation heavily depends on. It means that we now only update the encoder in the outer loop.) We empirically show that explicitly computing the reformulated losses of Eq. (7), Eq. (17) and Eq. (18) yield almost the same curves as MAML (with the EFIL assumption). Please note that the reformulated losses are used to update the encoders, for the linear layer $\mathbf{w}^0$, we explicitly update it using Eq. (5). Note that

although the performance models training using FOMAML, FOMAML with the zeroing trick, and SOMAML converge to similar testing accuracy, the overall testing performance during the training process is distinct. The results are averaged over three random seeds.

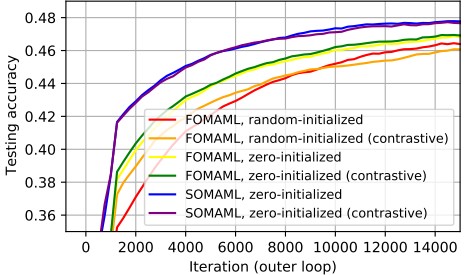

Figure 7: **Updating the encoder using the reformulated outer loop loss**. We experimentally validate that the testing accuracy of models trained using MAML (with no inner loop update of encoder) consists with that of models using their corresponding supervised contrastive losses, i.e., Eq. (7), Eq. (17) and Eq. (18).

### B.7    THE EFFECT OF INTERFERENCE TERM AND NOISY CONTRASTIVE TERM

Reformulating the loss of MAML into a noisy SCL form enables us to further investigate the effects brought by the interference term and the noisy contrastive term, which we presume both effects to be negative.

To investigate the effect of the interference term, we simply consider the loss adopted by first-order MAML as in Eq. (7) but with the interference term dropped (denoted as "$n_1 \times$"). As for the noisy contrastive term, the noise comes from the fact that "when the query and support data are in different classes, the sign of the contrastive coefficient can sometimes be negative", as being discussed in Section 2.4. To mitigate this noise, we consider the loss in Eq. (7) with the term $-(\sum_{j=1}^{N_{way}} q_j s_j) + s_u$ dropped from the contrastive coefficient, and denote it as "$n_2 \times$". On the other hand, we also implement a loss with "$n_1 \times, n_2 \times$", which is actually Eq. (9). We adopt the same experimental setting as Section B.6.

In Figure 8, we show the testing profiles of the original reformulated loss (i.e., the curve in red, labeled as "$n_1 \checkmark, n_2 \checkmark$"), dropping the interference term (i.e., the curve in orange, labeled as "$n_1 \times, n_2 \checkmark$"), dropping the noisy part of the contrastive term (i.e., the curve in green, labeled as "$n_1 \checkmark, n_2 \times$") or dropping both (i.e., the curve in blue, labeled as "$n_1 \times, n_2 \times$"). We can see that either dropping the interference term or dropping dropping the noisy part of contrastive coefficients yield profound benefit.

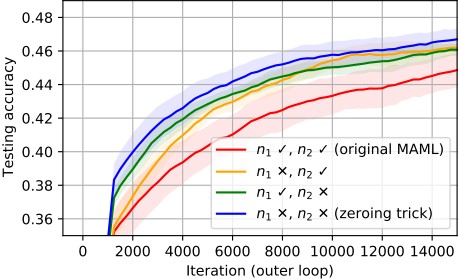

Figure 8: **The effect of the interference term and the noisy contrastive term.** We perform an ablation study of the reformulated loss in Eq. (7) by dropping the interference term (denoted as "$n_1$") or dropping the noisy part in the noisy contrastive term (marked as "$n_2$").

To better understand how noisy is the noisy contrastive term, i.e., how many times the sign of the contrastive coefficient is negative when the query and support data are in different classes, we explicitly record the ratio of the contrastive term being positive or negative. We adopt the same experimental setting as Section B.6.

The result is shown in Figure 9. When the zeroing trick is applied, the ratio of contrastive term being negative (shown as the red curve on the right subplot) is $0.2$, which is $\frac{1}{N_{way}}$ where $N_{way} = 5$ in our setting. On the other hand, when the zeroing trick is not applied, the ratio of contrastive term being negative (shown as the orange color on the right subplot) is larger than $0.2$. This additional experiment necessitates the application of the zeroing trick.

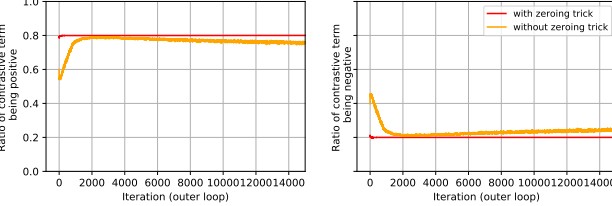

Figure 9: **The ratio of the contrastive term being positive or negative during training.** With the zeroing trick, the MAML becomes a SCL algorithm, so its ratio of negative contrastive term is $0.2$. The original MAML, however, is a noisy SCL algorithm, and thus its ratio of negative contrastive term is larger than $0.2$

## C   A GENERALIZATION OF OUR ANALYSIS

In this section, we derive a more general case of the encoder update in the outer loop. We consider drawing $N_{batch}$ tasks from the task distribution $D$ and having $N_{step}$ update steps in the inner loop while keeping the EFIL assumption.

To derive a more general case, we use $\mathbf{w_k}^{i,n}$ to denote the $k^{th}$ column of $\mathbf{w}^{i,n}$, where $\mathbf{w}^{i,n}$ is updated from $\mathbf{w^0}$ using support data $S_n$ for $i$ inner-loop steps. For simplicity, the $k^{th}$ channel softmax predictive output $\frac{\exp(\phi(s)^\top \mathbf{w_k}^{i,n})}{\sum_{j=1}^{N_{way}} \exp(\phi(s)^\top \mathbf{w_j}^{i,n})}$ of sample $s$ (using $\mathbf{w}^{i-1,n}$) is denoted as $\mathbf{s}_k^{i,n}$.

**Inner loop update for the linear layer** We yield the inner loop update for the final linear layer in Eq. (21) and Eq. (22).

$$\mathbf{w_k}^{i,n} = \mathbf{w_k}^{i-1,n} - \eta \frac{\partial L_{\{\varphi,\mathbf{w}^{i-1,n}\},S_n}}{\partial \mathbf{w_k}^{i-1,n}} = \mathbf{w_k}^{i-1,n} + \eta \underset{(s,t)\sim S_n}{\mathbf{E}} (1_{k=t} - \mathbf{s}_k^{i-1,n})\phi(s) \qquad (21)$$

$$\mathbf{w_k}^{N_{step},n} = \mathbf{w_k}^0 - \eta \sum_{i=1}^{N_{step}} \underset{(s,t)\sim S_n}{\mathbf{E}} (1_{k=t} - \mathbf{s}_k^{i-1,n})\phi(s) \qquad (22)$$

**Outer loop update for the linear layer** We derive the outer loop update for the linear layer in SOMAML, with denoting $I = \{1, 2, ..., N_{way}\}$:

$$
\begin{aligned}
\mathbf{w_k'}^0 &= \mathbf{w_k}^0 - \rho \sum_{n=1}^{N_{batch}} \frac{\partial L_{\{\varphi,\mathbf{w_k}^{N_{step},n}\},Q_n}}{\partial \mathbf{w_k}^0} \\
&= \mathbf{w_k}^0 - \rho \sum_{n=1}^{N_{batch}} \sum_{p_0=k,p_1\in I,...,p_{N_{way}}\in I} [(\prod_{i=0}^{N_{step}-1} \frac{\partial \mathbf{w_{P_{i+1}}}^{i+1,n}}{\partial \mathbf{w_{P_i}}^{i,n}}) \frac{\partial L_{\{\varphi,\mathbf{w}^{N_{step},n}\},Q_n}}{\partial \mathbf{w_{P_{N_{step}}}}^{N_{step},n}}]
\end{aligned}
\qquad (23)
$$

When it comes to FOMAML, we have

$$\mathbf{w_k'}^0 = \mathbf{w_k}^0 - \rho \sum_{n=1}^{N_{batch}} \frac{\partial L_{\{\varphi,\mathbf{w_k}^{N_{step},n}\},Q_n}}{\partial \mathbf{w_k}^{N_{step},n}} = \mathbf{w_k^0} + \rho \sum_{n=1}^{N_{batch}} \underset{(q,u)\sim Q_n}{\mathbf{E}} (1_{k=u} - \mathbf{q}_k^{N_{step},n})\phi(q) \qquad (24)$$

**Outer loop update for the encoder** We derive the outer loop update of the encoder under FOMAML as below. We consider the back-propagated error of the feature of one query data $(q, u) \sim Q_n$. Note that the third equality below holds by leveraging Eq. (21).

$$
\begin{aligned}
-\frac{\partial L_{\{\varphi,\mathbf{w}^{N_{step},n}\},Q_n}}{\partial \phi(q)} &= \mathbf{w_u}^{N_{step},n} - \sum_{i=1}^{N_{way}} (\mathbf{q}_i^{N_{step},n}\mathbf{w_i}^{N_{step},n}) \\
&= \sum_{i=1}^{N_{way}} (1_{i=u} - \mathbf{q}_i^{N_{step},n})\mathbf{w_i}^{N_{step},n} \\
&= \sum_{i=1}^{N_{way}} (1_{i=u} - \mathbf{q}_i^{N_{step},n})[\mathbf{w_i^0} + \eta \sum_{p=1}^{N_{step}} \underset{(s,t)\sim S_n}{\mathbf{E}} (1_{i=t} - \mathbf{s}_i^{p-1,n})\phi(s)] \\
&= \sum_{i=1}^{N_{way}} (1_{i=u} - \mathbf{q}_i^{N_{step},n})\mathbf{w_i^0} \\
&\quad + \eta \sum_{i=1}^{N_{way}} (1_{i=u} - \mathbf{q}_i^{N_{step},n}) \sum_{p=1}^{N_{step}} \underset{(s,t)\sim S_n}{\mathbf{E}} (1_{i=u} - \mathbf{s}_i^{p-1,n})\phi(s) \\
&= \sum_{i=1}^{N_{way}} (1_{i=u} - \mathbf{q}_i^{N_{step},n})\mathbf{w_i^0} \\
&\quad + \eta \underset{(s,t)\sim S_n}{\mathbf{E}} \sum_{p=1}^{N_{step}} [(\sum_{j=1}^{N_{way}} \mathbf{q}_j^{N_{step},n}\mathbf{s}_j^{p-1,n}) - \mathbf{s}_u^{p-1,n} - \mathbf{q}_t^{N_{step},n} + 1_{t=u}]\phi(s)
\end{aligned}
$$

$$\qquad (25)$$

**Reformulating the Outer Loop Loss for the Encoder as Noisy SCL Loss.** From Eq. (25), we can derive the generalized loss (of one query sample $(q, u) \sim Q_n$) that the encoder uses under FOMAML scheme.

$$
\begin{aligned}
L_{\{\varphi, \mathbf{w}^{N_{step}, n}\}, q} = & \sum_{i=1}^{N_{way}} \underbrace{(1_{i=u} - \mathbf{q}_i^{N_{step}, n}) \mathbf{w_i^0}^\top}_{\text{stop gradient}} \phi(q) \\
& + \eta \mathop{\mathbf{E}}_{(s,t) \sim S_n} \sum_{p=1}^{N_{step}} \underbrace{[(\sum_{j=1}^{N_{way}} \mathbf{q}_j^{N_{step}, n} \mathbf{s}_j^{p-1, n}) - \mathbf{s}_u^{p-1, n} - \mathbf{q}_t^{N_{step}, n} + 1_{t=u}] \phi(s)^\top}_{\text{stop gradient}} \phi(q)
\end{aligned}
\tag{26}
$$

# D    EXPERIMENTS ON MINI-IMAGENET DATASET

## D.1    EXPERIMENTAL DETAILS IN MINI-IMAGENET DATASET

The model architecture contains four basic blocks and one fully connected linear layer, where each block comprises a convolution layer with a kernel size of $3 \times 3$ and filter size of $64$, batch normalization, ReLU nonlineartity and $2 \times 2$ max-poling. The models are trained with the softmax cross entropy loss function using the Adam optimizer with an outer loop learning rate of 0.001 (Antoniou et al., 2019). The inner loop step size $\eta$ is set to 0.01. The models are trained for 30000 iterations (Raghu et al., 2020). The results are averaged over four random seeds, and we use the shaded region to indicate the standard deviation. Each experiment is run on either a single NVIDIA 1080-Ti or V100 GPU. The detailed implementation is based on Long (2018) (MIT License).

## D.2    THE EXPERIMENTAL RESULT OF SOMAML

The results with SOMAML are shown in Figure 10. Note that as it is possible that longer training can eventually overcome the noise factor and reach similar performance as the zeroing trick, the benefit of the zeroing trick is best seen at the observed faster convergence results when compared to vanilla MAML.

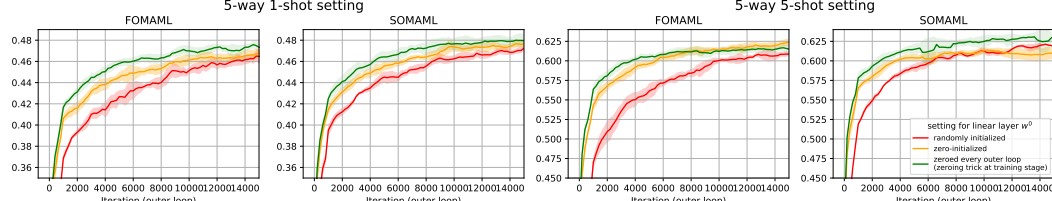

Figure 10: **Both FOMAML and SOMAML benefit from the zeroing trick**. We examine if reducing or removing the interference can increase the testing performance in models trained with FOMAML and SOMAML. The results suggest that SOMAML also suffers from the interference term. Note that the second subplots from the right shows lower testing performance of models trained with the zeroing trick as compared to the zero-initialized model. This may result from the overfitting problem. The curves in red: models trained with original MAML. The curve in orange: $\mathbf{w}^0$ is zero-initialized. The curve in green: models trained with the zeroing trick.

## D.3    COSINE SIMILARITY ANALYSIS ON SEMANTICALLY SIMILAR CLASSES VERIFIES THE IMPLICIT CONTRASTIVENESS IN MAML

In Figure 2, we randomly sample five classes of images under each random seed. Given the rich diversity of the classes in mini-ImageNet, we can consider that the five selected classes as semantically dissimilar or independent for each random seed. Here, we also provide the experimental outcomes using a dataset composed of five semantically similar classes selected from the mini-ImageNet dataset: French bulldog, Saluki, Walker hound, African hunting dog, and Golden retriever. Likewise to the original setting, we train the model using FOMAML and average the results over ten random seeds. As shown in Figure 11, the result is consistent with Figure 2. In conclusion, we

show that the supervised contrastiveness is manifested with the application of the zeroing trick even if a semantically similar dataset is considered.

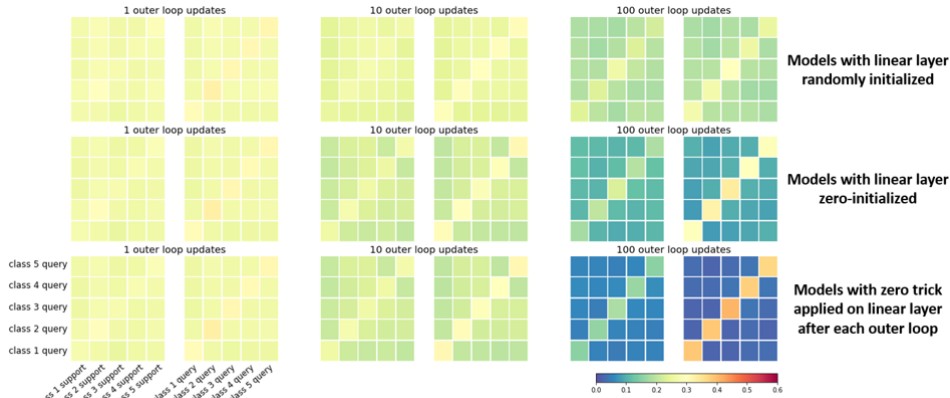

Figure 11: **The supervised contrastiveness is verified even using dataset composing of semantically similar classes of images.** Considering a dataset composing of different species of dogs, we again observe the tendency that the supervised contrastiveness is manifested when we zero-initialize the linear weight and apply the zeroing trick.

### D.4 EXPERIMENTAL RESULTS ON LARGER NUMBER OF SHOTS

To empirically verify if our theoretical derivation generalizes to the setting where the number of shots is large, we conduct experiment of a 5-way 25-shot classification task using FOMAML with four random seeds where we adopt mini-ImageNet as the example dataset. As shown in Figure 12, we observe that models trained with the zeroing trick again yield the best performance, consistent with our theoretical work that MAML with the zeroing trick is SCL without noises and interference.

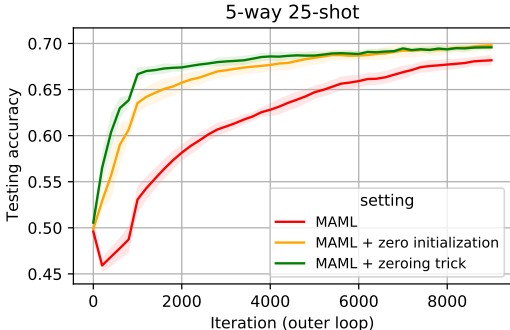

Figure 12: **The zeroing trick works when it comes to larger number of shots.** To examine if our work generalized to the scenario when the number of shots increases, we perform a 5-way 25-shot classification task. The result agrees with our results of 5-way 1-shot and 5-way 5-shot.

### D.5 THE ZEROING TRICK MITIGATES THE CHANNEL MEMORIZATION PROBLEM

The channel memorization problem (Jamal & Qi, 2019; Rajendran et al., 2020) is a known issue occurring in a non-mutually-exclusive task setting, e.g., the task-specific class-to-label is not randomly assigned, and thus the label can be inferred from the query data alone (Yin et al., 2020). Consider a 5-way K-shot experiment where the total number of training classes is $5 \times L$. Now we construct tasks by assigning the label $t$ to a class sampled from class $tL$ to $(t+1)L$. It is conceivable that the model will learn to directly map the query data to the label without using the information of the support data and thus fails to generalize to unseen tasks. This phenomenon can be explained from the perspective that the $t^{th}$ column of the final linear layer already accumulates the query features from $tL^{th}$ to $(t+1)L^{th}$ classes. Zeroing the final linear layer implicitly forces the model to use the imprinted information from the support features for inferring the label and thus mitigates

this problem. We use the mini-ImageNet dataset and consider the case of L = 12. As shown in Figure 13, the zeroing trick prevents the model from the channel memorization problem whereas zero-initialization of the linear layer only works out at the beginning. Besides, the performance of models trained with the zeroing trick under this non-mutually-exclusive task setting equals the ones under the conventional few-shot setting as shown in Figure 5. As the zeroing trick clears out the final linear layer and equalizes the value of logits, our result essentially accords with Jamal & Qi (2019) that proposes a regularizer to maximize the entropy of prediction of the meta-initialized model.

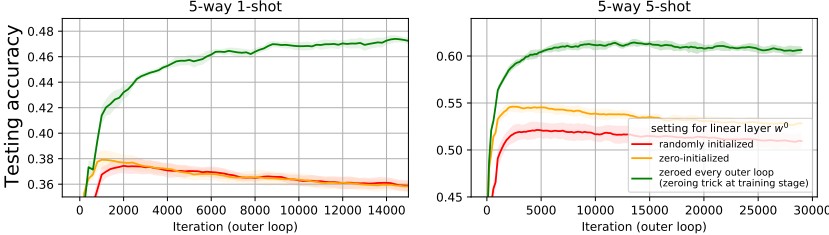

Figure 13: **The performance of the models trained on non-mutually exclusive tasks.** The models are trained under a non-mutually exclusive tasks setting where there is a one-to-one assignment between class and channel. Under this circumstance, the zeroing trick tackles the channel memorization problem and yields a performance similar to conventional few-shot settings.

## E   EXPERIMENTS ON OMNIGLOT DATASET

Omniglot is a hand-written character dataset containing 1623 character classes, each with 20 drawn samples from different people (Lake et al., 2015). The dataset set is splitted into training (1028 classes), validation (172 classes) and testing (423 classes) sets (Vinyals et al., 2016). Since we follow Finn et al. (2017) for setting hyperparamters, we do not use the the validation data. The character images are resized to $28 \times 28$. For all our experiments, we adopt two experimental settings: 5-way 1-shot and 5-way 5-shot where the batch size $N_{batch}$ is 32 and $N_{query}$ is 15 for both cases (Finn et al., 2017). The inner loop learning rate $\eta$ is 0.4. The models are trained for 3000 iterations using FOMAML or SOMAML. The few-shot classification accuracy is calculated by averaging the results over 1000 tasks in the test stage. The model architecture follows the architecture used to train on mini-ImageNet, but we substitute the convolution with max-pooling with strided convolution operation as in Finn et al. (2017). The loss function, optimizer, and outer loop learning rate are the same as those used in the experiments on mini-ImageNet. Each experiment is run on either a single NVIDIA 1080-Ti. The results are averaged over four random seeds, and the standard deviation is illustrated with the shaded region. The models are trained using FOMAML unless stated otherwise. The detailed implementation is based on Deleu (2020) (MIT License).

We revisit the application of the zeroing trick at the testing stage on Omniglot in Figure 14 and observe the increasing testing accuracy, in which such results are compatible with the ones on mini-ImageNet (cf. Figure 3 in the main manuscript). In the following experiments, we evaluate the testing performance only after applying the zeroing trick.

In Figure 15, the distinction between the performance of models trained with the zeroing trick and zero-initialized models is prominent, sharing remarkable similarity with the results in mini-ImageNet (cf. Figure 5 in the main manuscript) in both 5-way 1-shot and 5-way 5-shot settings. We also show the testing performance of models trained using SOMAML in Figure 16 under a 5-way 5-shot setting, where there is little distinction in performance (in comparison to the results on mini-ImageNet, cf. Figure 10 in the main manuscript) between the models trained with the zeroing trick and the ones trained with random initialization.

For channel memorization task, we construct non-mutually-exclusive training tasks by assigning the label $t$ (where $1 \le t \le 5$ in a few-shot 5-way setting) to a class sampled from class $t$L to $(t+1)$L where L is 205 on Omniglot. The class-to-channel assignment is not applied to the testing tasks. The result is shown in Figure 17. For a detailed discussion, please refer to Section D.5.

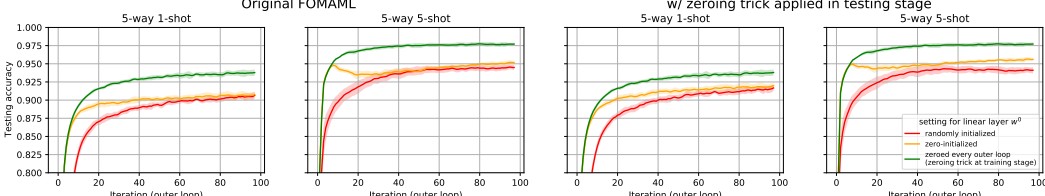

Figure 14: **Zeroing the final linear layer before testing time improves the testing accuracy on Omniglot.** The two subplots on the left: original testing setting. The two subplots at the right: the final linear layer is zeroed before testing time. The curves in red: the models whose linear layer is randomly initialized. The curves in yellow: the models whose linear layer is zeroed at initialization. The curves in green: the models whose linear layer is zeroed after each outer loop at training stage.

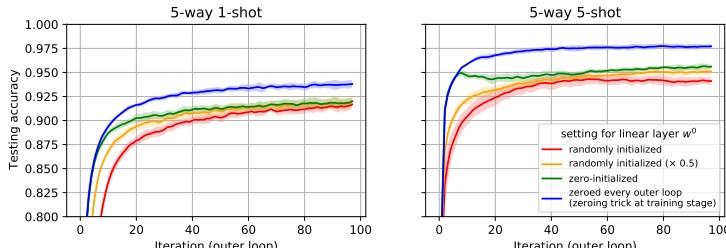

Figure 15: **Effect of initialization and the zeroing trick in testing accuracy on Omniglot.** The test performance of the models with reducing the initial norm of the weights of final linear layer is similar to that with the final linear layer being zero-initialized. The distinction in performance between models trained using the zeroing trick and zero-initialized model is more prominent in 5-way 5-shot setting.

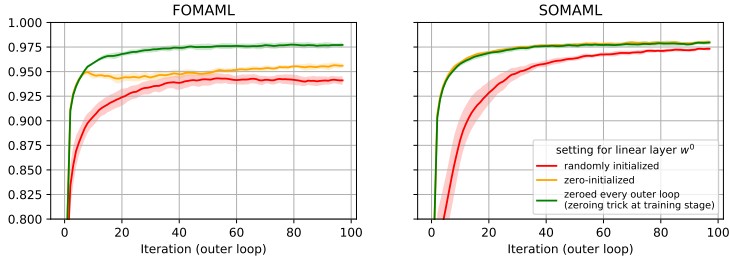

Figure 16: **The effect of the zeroing trick on models trained using FOMAML and SOMAML on Omniglot**. The results suggest that both zero-initialization and zeroing trick mitigate the interference empirically.

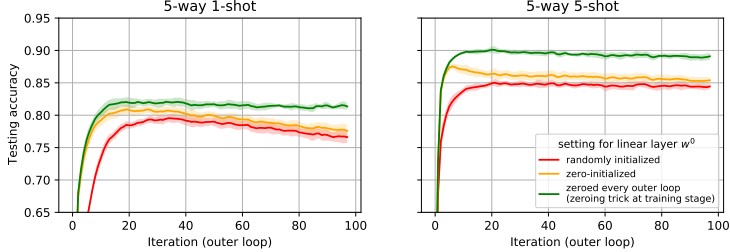

Figure 17: **The performance of the models trained on non-mutually exclusive tasks on Omniglot.** The results are compatible to those on mini-ImageNet (cf. Figure 13 in the main manuscript), suggesting that the zeroing trick alleviates the channel memorization problem.

