# OpenReview forum: "MAML is a Noisy Contrastive Learner in Classification"
_ICLR.cc/2022/Conference — ICLR 2022 Poster_

### Official Review · Reviewer_S6WC · 2021-11-02

**Correctness:** 4
**Technical Novelty And Significance:** 3
**Empirical Novelty And Significance:** 3
**Recommendation:** 8
**Confidence:** 3

**Main Review:**

I like the simplicity of the zeroing trick, and I found it useful to see it motivated via your theoretical arguments / perform well in your numerical experiments.

In my view, Section 2.1 could have done a better job of motivating why being a supervised contrastive learner is so appealing. This seems critical given that the rest of the paper provides conditions and theoretical/numerical results supporting that MAML is a (noisy) contrastive learner under certain conditions.

Your theoretical analysis focuses on a specific setting: few-shot classification tasks, softmax output, frozen convolutional encoder during inner loop To what extent do your findings generalize beyond these settings? I had been hoping that the discussion in Section 2.6 ("Generalization of our Analysis") would shed some light on this, but it only seemed to address one specific aspect of your analysis, namely the fact that you took $N_{batch}=N_{step}=1$.

Along the lines of the above, I think that the title and abstract are overly general (neither explicitly referencing your focus on few-shot classification problems, for example) given what you actually showed in the paper.

Sorry if I missed this, but did you freeze the encoder in the inner loop for your simulations? If so, did you also evaluate what happens when you didn't freeze it? This would help to give some experimental indication of how general your theoretical findings might be.

**Summary Of The Paper:**

This work shows that, in the setting of few-shot classification, MAML is a (noisy) contrastive learner. Complementary theoretical and experimental results are provided. The theoretical results also lead to a (to my knowledge) new proposal, namely to zero out the linear layer after each MAML outer loop update. In their experiments, the authors show that making this small change to the MAML algorithm can lead to meaningful improvements in performance.

**Summary Of The Review:**

I found the proposed zeroing trick (and the supporting theory/experiments) to be interesting. I would, however, encourage the authors to work to better motivate their study in the intro of the paper.

---

> ### Author Response · Authors · 2021-11-19
> **Response to Reviewer S6WC (1/2)**
>
> **Q1: Section 2.1 could have done a better job of motivating why being a supervised contrastive learner is so appealing. This seems critical given that the rest of the paper provides conditions and theoretical/numerical results supporting that MAML is a (noisy) contrastive learner under certain conditions.**
>
> A1: We thank the reviewer for the great suggestion. Supervised contrastive learning, proposed by [R1], is a generalization of several metric learning algorithms, such as triplet loss and N-pair loss. And these algorithms are well-known since they have been empirically and theoretically shown to enable models to learn more generalized representations. For example, in Table 2 of [R1], the authors show that SCL yields the best performance compared to SimCLR and CrossEntropy. Thus, by connecting MAML and SCL, we are attributing the success of MAML to SCL’s capacity in learning good representations with a formal theoretical proof. We added some text to our manuscript in Section 2.1 to better motivate the readers.
>
> [R1]. Supervised Contrastive Learning.
>
> **Q2: Your theoretical analysis focuses on a specific setting: few-shot classification tasks, softmax output, frozen convolutional encoder during the inner loop. To what extent do your findings generalize beyond these settings? I had been hoping that the discussion in Section 2.6 (“Generalization of our Analysis”) would shed some light on this, but it only seemed to address one specific aspect of your analysis.**
>
> We thank the reviewer for the insightful comment about the potential constraint. But we disagree that it is a specific setting. **Our main disagreement lies in that we are following the original MAML algorithm to establish our theoretical argument, which adopts softmax and is commonly used for few-shot classification.** Thus, it should not be considered as a limitation of our theoretical analysis. We detail them below.
>
> First of all, we would like to set the common ground for discussion by defining the term “few-shot learning”. In a typical $N$-way $K$-shot classification setting, one first present the $N$ novel classes of images with $K$ images per class to the model and then ask the model to classify $N$ novel classes with $J$ images per class. We call the presented $N\times K$ data the support data (refer to $S$ in our work) and $N\times J$ data the query data (refer to $Q$ in our work). Conventionally, we call it 1-shot learning if K=1, few-shot learning with K=1~5, and "low"-shot learning if K is sufficiently small. With this definition, we do not think it is necessary to mention “few-shot learning in our title” because our theoretical result (Theorem 1 and Corollary 1) applies to any number of data samples instead of “few-shot” samples. Therefore, the SCL view holds for the considered MAML algorithms in classification under the EFIL assumption (for EFIL, please refer to the footnote below this answer), including the few-shot learning settings.
>
> Second, **we disagree with the comment that “theoretical analysis focuses on a specific setting: few-shot classification.”** As is strongly emphasized in our manuscript, our work lies in understanding the working mechanism of MAML, and we believe our work is sufficient to achieve this goal. As MAML is an algorithm specifically designed to solve problems in the few-shot learning paradigm, it makes sense that we adopt few-shot learning as our numerical example setting. But we remark that our theoretical analysis that MAML is SCL under the EFIL assumption in classification holds beyond few-shot settings.
>
> Third, **our work is not limited to the LogSoftmax objective function.** As shown in Figure 1 of our manuscript, where we use the mean square error as our objective function, we get the same result: MAML is a supervised contrastive learning algorithm. The reason we use LogSoftmax in Section 2 is that this objective is the default setting of MAML. MAML is still an SCL algorithm even if we change the objective function. But, to our knowledge, we do not see any papers using different inner loop and outer loop objective functions. We believe this can be a new future research direction for the community.
>
> Last but not least, **we agree that our theoretical analysis relies on the EFIL assumption, but many existing and popular MAML algorithms, such as ANIL, MetaOptNet [R2], or R2-D2 [R3], leave the encoder unchanged and only change the linear weight during the inner loop and yield comparable results.** As a result, our EFIL assumption is empirically acceptable, and we believe the EFIL assumption will become more popular in a few years.
>
> We thank the reviewer for the constructive suggestions, and we add the discussion about the generalization of our work in Section 2.6.
>
> PS. For convenience, we call the assumption that “the Encoder is Frozen during the Inner Loop” as "EFIL".
>
> [R2] Meta-Learning with Differentiable Convex Optimization
>
> [R3] Meta-learning with differentiable closed-form solvers

---

> ### Author Response · Authors · 2021-11-19
> **Response to Reviewer S6WC (2/2)**
>
> **Q3: Along the lines of the above, I think that the title and abstract are overly general (neither explicitly referencing your focus on few-shot classification problems, for example) given what you actually showed in the paper.**
>
> A3: We thank the reviewer for the critical comment and the constructive suggestion. Because of the reason presented in A2 of our responses to you, we changed our title to “MAML is a noisy contrastive learner in classification” to make it more appropriate, as the reviewer kindly suggested.
>
>
>
> **Q4: Sorry if I missed this, but did you freeze the encoder in the inner loop for your simulations? If so, did you also evaluate what happens when you don't freeze it? This would help to give some experimental indication of how general your theoretical findings might be.**
>
> A4: We thank the reviewer for the comment and will add additional text to avoid vagueness. We did not freeze the encoder in the inner loop for all the experiments except in Appendix B.6 and B.7. Our experiments are designed to show how the zeroing trick benefits the original MAML algorithm by enhancing the supervised contrastiveness. As a result, we do not freeze the encoder during the inner loop. However, in Appendix B.7., we implement the reformulated loss (i.e., freezing the encoding layer in the inner loop) to see if the testing performance increases if we remove one of the interference terms. The resulting empirical outcomes fit our derivation very well.

---

> ### Comment · Reviewer_S6WC · 2021-11-30
> **Thanks for your replies**
>
> Based on your replies, I've raised my recommendation to Accept.

---

> > ### Author Response · Authors · 2021-11-30
> > **Thank you for increasing review score**
> >
> > We thank the reviewer for increasing the review score. Your review comments and suggestions have greatly improved the presentation of this work.
> >
> > Sincerely,
> >
> > Authors

---

### Official Review · Reviewer_DKb7 · 2021-11-03

**Correctness:** 3
**Technical Novelty And Significance:** 2
**Empirical Novelty And Significance:** 2
**Recommendation:** 5
**Confidence:** 4

**Main Review:**

Strengths:
- The SCL view proposed in the paper shows the link between MAML and the metric-based approaches for few-shot learning, such as matching network and prototypical network. In my view, this is interesting and might inspire future improvement of MAML for few-shot learning.

Weaknesses:
- The analysis in the paper is restricted. The SCL view seems to be only valid for few-shot learning. It is also over-simplified to assume that the inner-loop update does not affect the feature backbone if the task is not few-shot learning. I think a better title for the paper is "MAML is approximately a noisy contrastive learner for few-shot learning".

- There are no comparisons of MAML with the zeroing trick to metric-based few-shot learning methods in the experiments.  In my view, the SCL view of MAML indeed says that MAML is similar to metric-based approaches, especially when the zeroing trick is applied. The difference only lies in that metric-based approaches explicitly introduce metrics, while MAML with zeroing trick does perceptron-like learning on the top layer. I am surprised to see that the metric-based approaches are not even mentioned in the paper. It would make the results in the paper more insightful if this connection is explored in depth.

**Summary Of The Paper:**

In this paper, a new view of MAML under few-shot learning is proposed. The main result is that under the assumption that the inner loop updates are only applied on the top linear layer, MAML actually performs supervised contrastive learning (SCL). SCL shows that MAML learns the feature transformation that makes the intra-class feature distances small, meanwhile the inter-class feature distances large. The zeroing trick is proposed based on this result, showing performance gain in the experiments.



**Summary Of The Review:**

To summarize, I think the SCL view proposed in the paper would be interesting for researchers of few-shot learning. While the result is restricted to few-shot learning, meanwhile no exploration is done to analyse the connection between MAML and metric-based approaches. These drawbacks make the paper less insightful as I expected.

---

> ### Author Response · Authors · 2021-11-19
> **Response to Reviewer DKb7 (1/2)**
>
> **Q1-1: The analysis in the paper is restricted. The SCL view seems to be only valid for few-shot learning. ... I think a better title for the paper is “MAML is approximately a noisy contrastive learner for few-shot learning.”**
>
> A1-1. We thank the reviewer for the critical comment. We would like to present some inconsistencies between the reviewer’s comments and our main ideas.
>
> First of all, **we would like to clearly define the term “few-shot learning” to set the common ground for discussion**. In a typical $N$-way $K$-shot classification setting, one would like to first present the $N$ novel classes of images with $K$ images per class to the model and then ask the model to classify $N$ novel classes with $J$ images per class. Basically, we call the presented $N\times K$ data the support data (refer to $S$ in our mathematical derivation) and $N\times J$ data the query data (refer to $Q$ in our derivation). Conventionally, we call it 1-shot learning if $K$=1, few-shot learning with $K$=1~5, and "low"-shot learning if K is sufficiently small. Based on this definition, we do not think it is necessary to mention “few-shot learning in our title” because our theoretical result (Theorem 1 and Corollary 1) applies to any number of data samples instead of “few-shot” samples. Therefore, the SCL view holds for the considered MAML algorithms in classification under the EFIL assumption (for EFIL, please refer to the footnote below this answer), including the few-shot learning settings. We here provide the empirical results of the 5-way 25-shots setting which is not a “few-shot” setting (https://imgur.com/a/Ow0PEAC). And we can clearly see that even for a large-shot setting, the zeroing trick still works. We added the result to our manuscript in Section D.4.
>
> Second, **we disagree with the comment that “The analysis in the paper is restricted.”** Our analysis is as general as the considered MAML algorithms in classification and extends to the applications therein, including few-shot learning settings. We believe the perceived restriction is actually on the methodology of MAML itself, rather than our analysis.  As is strongly emphasized in our manuscript, our work lies in understanding the working mechanism of MAML, and we believe our work is sufficiently general to achieve this goal. Thus, it makes sense that we adopt few-shot learning as our numerical example setting (but our theoretical analysis is not limited to few-shot settings) and conclude that MAML is SCL under the EFIL assumption.
>
> Last but not least, **we admit a limitation of our analysis in the classification setting, because SCL is not properly defined for non-classification tasks**. Thus we decided to change our title to "MAML is a Noisy Contrastive Learning in Classification." to reflect this limitation.
>
> PS. For convenience, we call the assumption that “the Encoder is Frozen during the Inner Loop” as "EFIL".
>
> **Q1-2. It is also over-simplified to assume that the inner-loop update does not affect the feature backbone if the task is not few-shot learning.**
>
> A1-2: We thank the reviewer for the critical comment. In [R1], where the ANIL algorithm (please refer to foot note for ANIL) is proposed, they show the empirical results that are not confined to few-shot learning. For example, In figure 2 (https://i.imgur.com/tkvWrVa.png) and figure 3 (https://i.imgur.com/xGyYQQe.png) of [R1], the authors show that, during the inner loop, the representations of the intermediate layers only change slightly. These results are related to “the inner loop update” and are independent of “few-shot learning”, indicating that the EFIL assumption (for EFIL, please refer to the footnote) is acceptable where there is a nested-loop scheme. As a result, the EFIL assumption stands empirically and is acceptable even for the non-few-shot scenarios.
>
> PS. For convenience, we call the assumption that the encoder is not updated in the inner loop adopted in [R1] as “ANIL", which stands for “Almost No Inner Loop”
>
> PS. For convenience, we call the assumption that “the Encoder is Frozen during the Inner Loop” as "EFIL".
>
> [R1]. Rapid Learning or Feature Reuse? Towards Understanding the Effectiveness of MAML

---

> > ### Comment · Reviewer_DKb7 · 2021-11-29
> > **thanks for the responses**
> >
> > Dear authors,
> >
> > Thanks for the responses. I greatly appreciate the effort being made on revising the paper. For me, it is nice to see that discussions about the relationship to metric-based few-shot learning are added. I think the paper provides an interesting view of MAML. But for making the results more significant, I think experimental comparisons to metric-based few-shot learning methods are necessary to be included. For this reason, I will keep my evaluations unchanged currently.

---

> > > ### Author Response · Authors · 2021-11-30
> > > **Thank you for your feedback; we would like to make a point**
> > >
> > > We thank the reviewer for the post-rebuttal feedback. We are also glad to know that the reviewer finds our responses useful. We understood that the reviewer wanted to see experimental comparisons to metric-based few-shot learning methods. However, conducting such experiments does not help illustrate the intrinsic contrastive property of MAML empirically, nor does it strengthen the connection between MAML and metric-based few-shot learning. Furthermore, we argue that having such empirical comparison would not change the two main conclusions and contributions of this work: (1) The theoretical analysis that MAML in classification under the EFIL assumption follows noisy supervised contrastive learning; (2) The use of zeroing trick can improve the learning performance of the considered MAML algorithms.
> > >
> > > We hope that our clarification would help the reviewer understand that the suggested comparison would not affect our results and findings, and therefore it is not necessary to be included.
> > >
> > > Sincerely,
> > >
> > > Authors

---

> ### Author Response · Authors · 2021-11-19
> **Response to Reviewer DKb7 (2/2)**
>
> **Q2: There are no comparisons of MAML with the zeroing trick to metric-based few-shot learning methods in the experiments. ... It would make the results in the paper more insightful if this connection is explored in depth.**
>
> A2: We thank the reviewer for the suggestion. We agree that under the assumption, MAML is approximately a metric-based few-shot learning algorithm with a metric of inner product and with the prototypes being the weighted sum of the support features. From a high-level perspective, second-order MAML is similar to some famous metric-based few-shot learning algorithms, such as MatchingNet, ProtoNet, and RelationNet, under the EFIL assumption. The main differences lie in the metric and the ways prototypes are constructed. Therefore, we accept the reviewer’s suggestion and strengthen the relationship between MAML and metric-based few-shot learning algorithms in Section 2.6.

---

### Official Review · Reviewer_XCch · 2021-11-04

**Correctness:** 3
**Technical Novelty And Significance:** 3
**Empirical Novelty And Significance:** 3
**Recommendation:** 5
**Confidence:** 4

**Main Review:**

Strengths
=======

The analysis is quite interesting. Their efforts make it explicit the interaction between the support and query set in MAML, and how MAML learns feature encoding. They discover a noisy supervised contrastive loss term in the outer loop loss using the support features as positive and negative samples. Interestingly, they discover another interference term in the outer loop loss which may degrade performance of MAML if last layer linear weights are randomly initialized.

To remove the interference term, and also to remove the noise in the supervised contrastive loss, they propose a simple zeroing trick to set the initial weights to be zero after each outer loop update. The trick is simple and reasonable given their analysis results.

They conduct experiments to verify the contrastiveness in MAML and improvements using the zeroing trick.

Overall, the paper is quite clear and easy to follow.

Question and weakness
===================

Comparison between eqn (7) and (8) is not clear. Can the authors discuss further the different range of stop gradient?

The results in Figure 2 are nice but limited, not sure if these can be observed for other classes. What are the 5 classes in the experiment? Can the authors experiment other classes, especially some semantically similar classes?

Further to the analysis in Figure 2, I am not sure if they really "verify the supervised contrastiveness". In particular, the results show that support and query features of same classes become similar as training progresses. I do not think this is particular for supervised contrastiveness. For example, authors can try standard transfer learning and fine-tuning and see if such pattern of consine similarities can be observed.

I am ok with the assumption of freezing the encoder in the inner loop and update only linear layer. But with that, I thought some remarks of the paper sound trivial. For example, “In the inner loop, the features of support data are preserved in the linear layer via inner loop update. In the outer loop, the softmax output of the query data thus contains the inner products between the support features and the query feature.” If you can update only the last linear layer in the inner loop then of course support data feature can only reside there, and therefore in the outer loop it would be inner products between support features and query input as the last layer is a linear layer. Did I miss anything?

The analysis focuses on classification. Inner product and softmax are critical components in their analysis. But MAML has been applied to other problems, e.g. regression. Perhaps the focus on classification should be reflected in the paper title.



**Summary Of The Paper:**

The paper analyzes MAML algorithms. Assuming in the inner loop the encoder is fixed and only last linear layer is updated, they analyze the gradient update and loss terms in the inner loop and outer loops (sec 2.3). Through this effort, the authors claim that there are noisy supervised contrastive term in the outer loop loss (eqn (7) and (8)). They further claim that there are additional interference terms which may degrade the performance of MAML at the beginning of training when the linear layer weights are largely random. To overcome this, they propose a simple zeroing trick by zeroing the initial linear layer weights after each outer loop update, essentially removing the interference terms (eqn (9) and (10)). They conduct experiments to support the contrastiveness in MAML and performance improvement using zeroing trick.

**Summary Of The Review:**

Several concerns are listed above. I would be happy to re-consider my recommendation if the responses are reasonable.

---

> ### Author Response · Authors · 2021-11-19
> **Response to Reviewer XCch (1/3)**
>
> **Q1. The comparison between Eq.(7) and (8) is not clear. Can the authors discuss further the different range of stop gradients?**
>
> A1. Thank you for giving us a chance to elaborate on the difference between different ranges of gradient stopping.
>
> To be self-contained, we show Eq.(9) and (10) below. For simplicity, we use $\tilde{L}$ to represent $L_{\{\varphi,\mathbf{w}^1\},q}$ and $C$ to represent the contrastive coefficient $[-\sum_{j=1}^{N_{way}} \text{q}_j \text{s}_j+\text{s}_u+\text{q}_t-1_\mathrm{t=u}]$, which depends on the support data, query data and their labels.
>
> Eq.(9): $\tilde{L}=\sum_{j=1}^{N_{way}} \underbrace{(\text{q}_j-1_\mathrm{j=u})\mathbf{w_j}^{0\top}}_\{\text{stop gradient}}\phi(q) + \eta \mathop{\mathbf{E}}_\{(s,t)\sim S} \underbrace{C\phi(s)^\top}_\{\text{stop gradient}}\phi(q)$
>
> Eq.(10): $\tilde{L}=\sum_\{j=1}^{N_{way}}\underbrace{(\text{q}_j-1_\mathrm{j=u})\mathbf{w_j}^{0\top}}_\{\text{stop gradient}}\phi(q)+\eta \mathop{\mathbf{E}}_\{(s,t)\sim S} \underbrace{C}_\{\text{stop gradient}}\phi(s)^\top\phi(q)$.
>
> As we aim to understand the effect of different ranges of gradient stop, we neglect the first term in both equations. Therefore, we have:
>
> Eq.(9-1): $\tilde{L} =  \eta\mathop{\mathbf{E}}_\{(s,t)\sim S} \underbrace{C\phi(s)^\top}_\{\text{stop gradient}}\phi(q)$.
>
> Eq.(10-1): $\tilde{L} =  \eta \mathop{\mathbf{E}}_\{(s,t)\sim S} \underbrace{C}_\{\text{stop gradient}}\phi(s)^\top\phi(q)$.
>
> Since MAML utilizes these losses to update the encoder $\phi$, by product rule we have $\frac{\partial \tilde{L}}{\partial\varphi} = \mathop{\mathbf{E}}_\{(q,u)\sim Q}\frac{\partial \tilde{L}}{\partial\phi(q)}\frac{\partial \phi(q)}{\partial\varphi} + \mathop{\mathbf{E}}_\{(s,t)\sim S}\frac{\partial \tilde{L}}{\partial\phi(s)}\frac{\partial\phi(s)}{\partial\varphi}$. Speficially, to understand how the features are changed under these losses, we can look at the terms $\frac{\partial L}{\partial\phi(q)}$ and $\frac{\partial L}{\partial\phi(s)}$ by differentiating Eq.(9-1) and Eq.(10-1) with respect to the encoded feature of data as the following.
>
> Eq.(9-2): $\frac{\partial\tilde{L}}{\partial\phi(q)}=\eta\mathop{\mathbf{E}}_{(s,t)\sim S} C \phi(s)$.
>
> Eq.(9-3): $\frac{\partial\tilde{L}}{\partial\phi(s)}=0$.
>
> Eq.(10-2): $\frac{\partial\tilde{L}}{\partial\phi(q)}=\eta\mathop{\mathbf{E}}_{(s,t)\sim S}C\phi(s)$.
>
> Eq.(10-3): $\frac{\partial\tilde{L}}{\partial\phi(s)}=\eta\mathop{\mathbf{E}}_\{(s,t)\sim S}{C}\phi(q)$.
>
> Obviously, as Eq.(9-3) is zero, we know that in FOMAML, the update of the encoder does not aim to change the support features. Instead, in FOMAML, the encoder is updated so as to move the query features closer to support features of the same class and further to support features of different classes. On the contrary, we can tell from the above equations that in SOMAML, the encoder is updated to make support features and query features closer if both of them come from the same class, and to make support features and query features further if they come from different classes.
>
> We further illustrate the difference in this figure (https://imgur.com/a/8cniyGW). For simplicity, we do not consider the scale of the coefficients but their signs. The left subplot indicates that the FOMAML loss guides encoder to update so that the query feature 1) moves towards the support features of the same class and 2) against the support features of the different classes. On the other hand, the SOMAML loss guides the encoder to update so that 1) when the support data and query data belong to the same class, their features get closer, and 2) otherwise further. This explains why SOMAML typically converges faster than FOMAML. We add discussion and the illustration in Appendix B.5.
>
> **Q2: Not sure if the results in Figure 2 can be observed for other classes. What are the five classes in the experiment? Can the authors experiment with other classes, especially some semantically similar classes?**
>
> A2: We thank the reviewer for the questions. For the setting in Figure 2, we randomly sample five classes of images under ten random seeds. Given the rich diversity of the classes in Mini-ImageNet, the reviewer can consider the five selected classes as semantically dissimilar or independent for each random seed.
>
> To address the reviewer’s concern, we provide the experimental outcomes using a dataset composed of five semantically similar classes selected from the Mini-ImageNet dataset: French bulldog, Saluki, Walker hound, African hunting dog, and Golden retriever. Likewise to the original setting, we train the model using FOMAML and average the results over ten random seeds. As shown in the link (https://imgur.com/a/oyQWZ8k), the result is consistent with Figure 2 in our manuscript. In conclusion, we show that the supervised contrastiveness is manifested with the application of the zeroing trick even if a semantically similar dataset is considered. We added the result to our manuscript in Section 3.2 and Section D.3.

---

> ### Author Response · Authors · 2021-11-19
> **Response to Reviewer XCch (2/3)**
>
> **Q3: In Figure 2, I am not sure if they really “verify the supervised contrastiveness”. In particular, the results show that support and query features of the same classes become similar as training progresses. I do not think this is particularly for supervised contrastiveness.**
>
> A3: We thank the reviewer for the insightful and critical comment. We would like to emphasize that whether other machine learning schemes follow similar patterns or not should not affect the conclusion of our work. Our result shows both theoretically and empirically that the considered MAML algorithms in classification are implicitly adopting (noisy) supervised contrastiveness loss for learning. We agree with the reviewer that our result does not rule out the possibility of other machine learning schemes implicitly following the same methodology or obtaining similar results on supervised contrastivenss, and, to our best knowledge, these results may be empirically observed but may not be theoretically proven. If the reviewer feels necessary, we are willing to conduct similar experiments in transfer learning and fine-tuning, though the conclusion will be irrelevant to our claim. Please let us know.
>
> Here, we would like to emphasize the primary purpose of Section 3.2: From our derivation, we conceptually know that MAML is a noisy SCL. One way to validate our viewpoint is to show if the contrastiveness is manifested once we remove the noise terms. Therefore, we compare the original MAML algorithm with the other two conditions: one with the linear layer $w$ zero-initialized and another with the zeroing trick. And we find that the supervised contrastiveness dramatically increases. This helps us verify that MAML is a noisy contrastive learner. We highly suggest the reviewer look at Figure 2 again and compare the heatmaps between the different rows. We understand that our figure may mislead the readers to compare the heatmaps from column to column, and we add additional text to guide the readers in the caption of Figure 2.
>
> **Q4: I am ok with the assumption of freezing the encoder in the inner loop and updating only the linear layer. But with that, I thought some remarks of the paper sounded trivial. For example, “In the inner loop, the features of support data are preserved in the linear layer via inner loop update. In the outer loop, the softmax output of the query data thus contains the inner products between the support features and the query feature.”**
>
> A4: We thank the reviewer for pointing out this part. The reviewer correctly understands what we try to convey in this work. However, we would like to shift the reviewer’s attention to the originality of our work.
>
> First of all, these remarks might seem trivial and intuitive, but their observation and insights are not, especially in our findings that there are cross-task noisy terms hidden in native MAML algorithms. Here we would like to use [R1] to explain our intuition and novelty, from which we obtain our assumption from their proposed ENIL algorithm (for the term ENIL, please refer to the footnote). In the ENIL algorithm, the encoder is frozen during the inner loop, and the overall training speed notably increases with comparable results.  [R1] first appeared in Arxiv in September 2019 and has received 178 citations to date. Although [R1] provides intensive and extensive empirical evidence about the ANIL algorithm, we have not yet seen any work, including [R1] itself, to further expand from their empirical observation and theoretically prove the connection between meta-learning and contrastive learning. To our knowledge, our work is the first paper to provide a theoretical understanding of the working mechanism of MAML under this assumption. As the pioneering work to identify the importance of the insight hidden in [R1] and conceive up a relation that bridges MAML and supervised contrastive learning, we would argue that our work is far from being trivial.
>
> Secondly, we further improved the MAML algorithm in convergence rate by the proposal of the zeroing trick. In our analysis, MAML unfolds itself as an implicit SCL algorithm, but it also contains two interference terms that interfere with the convergence speed and final performance. Upon this theoretical observation, we propose the effective zeroing trick. We call it a “trick” because we understand this is simple. But it is by no means trivial as it requires insights into the assumption and the observation of noises. As a result, the remarks of our work may sound trivial, but they actually bridge us to the proposal of the zeroing trick that further improves MAML.
>
> PS. ANIL (almost no inner loop) algorithm refers to the MAML algorithm but with “freezing the encoder during the inner loop.”
>
> [R1]. Rapid Learning or Feature Reuse? Towards Understanding the Effectiveness of MAML

---

> ### Author Response · Authors · 2021-11-19
> **Response to Reviewer XCch (3/3)**
>
> **Q5: The analysis focuses on classification, Inner product and softmax are critical components in their analysis. But MAML has been applied to other problems, e.g., regression. So perhaps the focus on classification should be reflected in the paper title.**
>
> A5: We thank the reviewer for the insightful comment. We agree that our derivation focuses on the classification task. Therefore, we changed our title to “MAML is a noisy contrastive learner in classification” to make it more appropriate. However, we disagree with some of the mentioned limitations of our work. Our main disagreement lies in that we are following the model and setting that MAML uses, and thus this should not be considered as a limitation. We detail our points below.
>
> First, our work is not limited to the LogSoftmax objective function. As shown in the motivating example in Figure 1, where we use the mean square error (MSE) as our inner loop and outer loop objective function, we get the same result that MAML is a supervised contrastive learning algorithm. The reason we use LogSoftmax is that this objective is the default setting in MAML. Besides, we believe that future work can be enlightened to adopt different losses in the inner and outer loops, such as using MSE for the inner loop but LogSoftmax for the outer loop.
>
> Second, we consider the inner product as a measure of similarity in our analysis because the last operation of the model is to perform matrix multiplication between the encoded features and the weight of the linear layer, which is a convention of neural network design. With the insight that MAML is an SCL algorithm, we believe that future work may look forward to improving MAML by changing the last operation of the model to other operations, such as cosine similarity or negative Euclidean distance. However, even under these circumstances, we can still conclude that MAML is SCL in the classification tasks by simply changing the inner product part in our analysis to other similarity measurements.
>
> We argue that following the default setting of MAML should not be considered the constraint of our work. Besides, our work can be easily extended to other objective functions and operations. We thank the reviewer for the constructive suggestions, and we have added the discussion about the generalization of our work in Section 2.6.

---

> ### Author Response · Authors · 2021-11-30
> **Follow-up message for Reviewer XCch**
>
> Dear Reviewer XCch,
>
> Based on your initial review comment "*I would be happy to re-consider my recommendation if the responses are reasonable.*", we would like to follow up to remind the reviewer about our author responses and revised version, which incorporated all the suggested changes and review comments. We would like to use this opportunity to thank you for your valuable review comments, which have improved the presentation and further clarified the research contributions of this work.  We believe our revision and responses have addressed your concerns. Shall you have any further questions, please don't hesitate to let us know!
>
> Sincerely,
>
> Authors

---

### Author Response · Authors · 2021-11-22
**Looking forward to reviewer's feedback on our revised version and rebuttal**

Dear Area Chair and Reviewers,

Today (22th) is the deadline for allowing making further updates on our submission. While we are happy to continue to interact with the reviewers and area chair through 29th, we would like to ensure the current updated version is clear to reviewers and is helpful for addressing the reviewers' concerns and suggestions. We look forward to the post-rebuttal feedback and continued discussion from area chair and reviewers!

Sincerely,

Authors

---

> ### Author Response · Authors · 2021-11-29
> **Follow-up message from authors**
>
> Dear Area Chair and Reviewers,
>
> As the discussion deadline is closing soon, we would like to follow up to ensure we have successfully conveyed the merits and main contributions of our work. We took the silence of the post-rebuttal discussion as a positive sign indicating our revised version and responses had addressed your concerns. In the meantime, we are happy to answer any questions the AC and Reviewers may have. Please don't hesitate to let us know!
>
> Sincerely,
>
> Authors

---

### Decision · Program_Chairs · 2022-01-20

**Decision:**

Accept (Poster)

**Comment:**

This paper connects MAML to contrastive learning under some simplifying assumptions and with slight modifications in the setting.
Specifically, the authors show that if the inner loop updates are only applied on the top linear layer, MAML is equivalent to supervised contrastive learning (SCL). This means that MAML learns a feature transformation that brings in-class representations closer and representations across classes far away. The zeroing trick the authors propose seems to give some performance gain in the experiments and is an actionable insight from their theory.

Overall the paper is very interesting and (as far as I know) novel. The proposed zeroing trick is supported by theory and experiments and is justifies the previous theoretical narrative.

Some reviewers raised concerns on motivation and numerous clarification questions that the authors have addressed to a large extent in my opinion.